# Are We Learning Yet? A Meta-Review of Evaluation Failures Across Machine Learning

**Thomas I. Liao**
Scale AI
`thomas.liao@scale.com`

**Rohan Taori**
Stanford University
`rtaori@stanford.edu`

**Inioluwa Deborah Raji**
UC Berkeley
`rajiinio@berkeley.edu`

**Ludwig Schmidt**
Toyota Research Institute
University of Washington
`schmidt@cs.uw.edu`

## Abstract

Many subfields of machine learning share a common stumbling block: evaluation. Advances in machine learning often evaporate under closer scrutiny or turn out to be less widely applicable than originally hoped. We conduct a meta-review of 107 survey papers from computer vision, natural language processing, recommender systems, reinforcement learning, graph processing, metric learning, and more, organizing a wide range of surprisingly consistent critique into a concrete taxonomy of observed failure modes. Inspired by measurement and evaluation theory, we divide failure modes into two categories: internal and external validity. Internal validity pertains to evaluation on a learning problem in isolation, such as improper comparisons to baselines or overfitting from test set re-use. External validity relies on relationships between different learning problems, for instance, whether progress on a learning problem translates to progress on seemingly related tasks.

## 1 Introduction

Most empirical papers in machine learning follow the benchmarking paradigm for evaluation. There is a myriad of datasets and tasks in the literature, and what it means for a machine to "learn" has interpretations from mirroring human-like intelligence to solving a specific practical task. Nevertheless, whether a new method has merit is usually determined by evaluating a trained model on a held-out test set and comparing its performance to prior work. If the new model improves over the relevant baselines, the method represents an algorithmic contribution. Since the benchmark itself is often only a challenge problem specifically constructed for research, the underlying assumption is that the new method will also yield performance improvements on real-world problems similar to the benchmark.

Benchmarking was popularized in machine learning in the 1980s through the UCI dataset repository and challenges sponsored by DARPA and NIST [24, 35, 55, 81]. Since then, benchmark evaluations have become the core of most empirical machine learning papers. The impact of benchmarking is illustrated by the ImageNet competition [31, 131], which seeded much of the excitement in machine learning since 2010. Winning entries such as AlexNet [77] and ResNets [57] have become some of the most widely cited papers across all sciences.

Evaluating algorithmic progress with benchmarks is a double-edged sword. On the one hand, benchmarks come with a clearly defined performance metric that enables objective assessments of different algorithms. On the other hand, summarizing a new algorithm with a single performance number creates an illusion of simplicity that ignores the many underlying assumptions in the learning problem posed as a benchmark. Indeed, an increasing number of machine learning papers take a critical perspective on recent algorithmic advancements and find important flaws in current evaluation

35th Conference on Neural Information Processing Systems (NeurIPS 2021) Track on Datasets and Benchmarks.

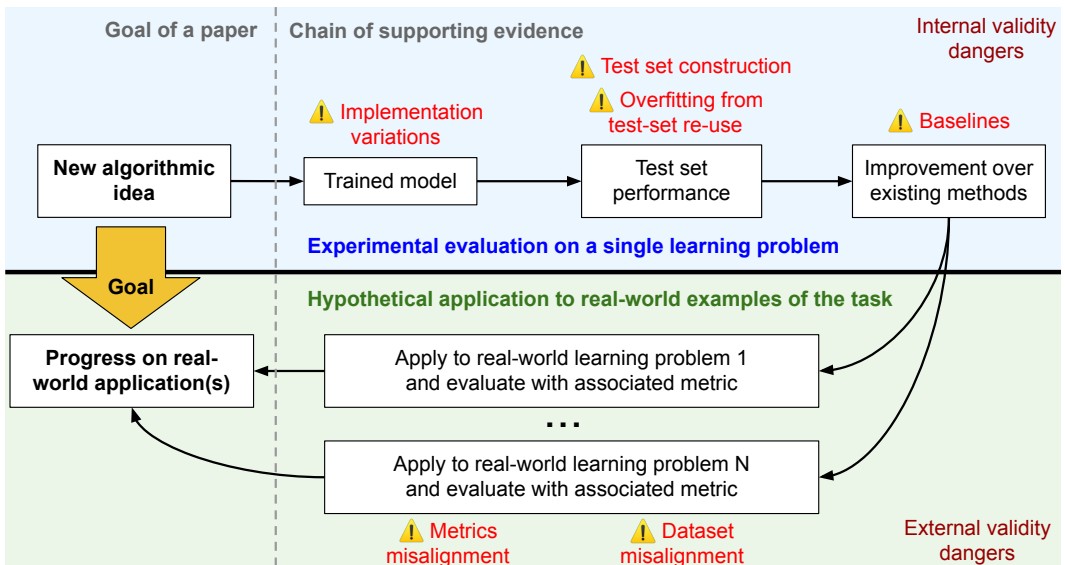

Figure 1: Our framework for benchmark-based evaluations of machine learning algorithms and associated validity concerns. In the benchmark paradigm, papers which propose a new algorithmic idea demonstrate its effectiveness by comparing to results of prior work on a specific learning problem (the benchmark). The underlying assumption is that the benchmark is representative for a broader task and hence the performance improvements will transfer to real-world applications. This chain of reasoning relies on multiple steps with various potential validity issues.

practices. For instance, most claimed advances from the past few years of recommender systems research failed to improve over established baselines and evaporate under closer scrutiny [25, 125]. Given the key role benchmarking plays in machine learning, such evaluation flaws threaten to undermine the perceived algorithmic gains in recent years.

In this paper, we provide a systematic taxonomy of failures in the benchmarking paradigm in order to put current evaluation practices on solid foundations. Our taxonomy draws from 107 analysis papers which study specific machine learning evaluations; we describe further how we arrived at this taxonomy in Appendix 6. Despite the diversity of tasks and algorithms, we find that the same evaluation failures repeat across diverse areas such as computer vision, natural language processing, recommender systems, reinforcement learning, graph processing, metric learning, and more. Based on lessons from evaluation theory [92], we divide the failure modes into two categories:

- **Internal validity** refers to issues that arise within the context of a single benchmark.
- **External validity** asks whether progress on a benchmark transfers to other problems.

Figure 1 illustrates our taxonomy of evaluation failures in machine learning. Our taxonomy can serve as a resource for machine learning researchers and practitioners to check for evaluation issues in their own disciplines. Since many failure modes occur in several fields, insights from one field will transfer to others. Additionally, our paper contributes insights to the ongoing discussion around evaluation practices in machine learning. Finally, our taxonomy of external validity criteria offers a starting point for research in this area. The relationships between different datasets and learning problems are not yet well understood; more work is needed to understand the scope of current benchmarks.

Next we introduce our framework for evaluation validity in machine learning, which organizes the common failures modes described in Sections 3 and 4. Section 5 then discusses limitations of the benchmarking paradigm itself before we conclude in Section 6. An overview of the papers that inform this survey can be found in Appendices D and E.

## 2   A conceptual framework for machine learning evaluations

Empirical machine learning evaluations are ultimately tied to datasets. A key question is to what extent the datasets used to measure algorithm performance (e.g., ImageNet [31, 131] or GLUE [158]) represent the problem a paper claims to address (e.g., image classification or natural language

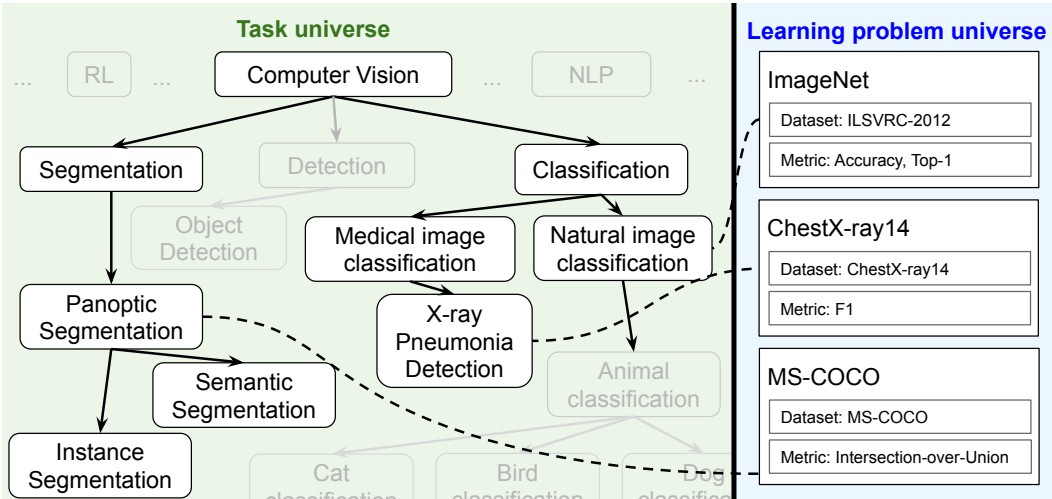

Figure 2: An example of a task hierarchy and associated learning problems. Tasks are abstract problem statements formulated independently from datasets and exist at various levels of granularity, giving rise to a hierarchy. In contrast, a learning problem combines a specific dataset and a particular metric to instantiate one or more tasks. Many learning problems can attempt to instantiate the same task, and the relationships between different learning problems is the focus of external validity.

understanding). To make this distinction clear from the beginning, we define two different kinds of problem statements. These two notions for "learning from data" distinguish between concrete problems defined via *datasets* and abstract problems defined via formal or informal *semantics*.

## 2.1 Two kinds of problem statements: learning problems vs. tasks

**Learning problems.** A learning problem comprises a dataset of (input, output) pairs and an associated evaluation metric for scoring proposed solutions (functions from the input to the output space). A learning problem is fully defined by these two parts and requires no further reference to external semantics or data; e.g., the ILSVRC-2012 dataset (ImageNet) with top-1 accuracy as metric.

**Tasks.** A task is a problem statement defined abstractly, either via natural language or in a formal way. A task does not necessarily have a single true definition and we do not aim to establish any task definitions. Tasks can exist at varying granularities, e.g., from "dog vs. cat classification" to "animal classification" to "image classification", which naturally gives rise to a hierarchy (see Figure 2). Tasks are omnipresent in the machine learning literature as a way to frame contributions. For the purpose of evaluation, tasks are usually instantiated by learning problems. As an example, MNIST, CIFAR-10, and ImageNet all instantiate the "image classification" task.

Given these definitions, a *benchmark* is a learning problem framed as an indicator of progress on some task. Benchmarks usually come with a leaderboard, competition, or other context that establishes the current state of the art. For example, improving accuracy on ImageNet can be considered as making improvements on the image classification task in the context of the ILSVRC competition [131].

## 2.2 Internal and external validity in machine learning evaluations

The distinction between learning problems and tasks also separates validity issues in machine learning into *internal* issues, i.e., issues arising within the context of a single learning problem, and *external* issues, i.e., issues stemming from the relationship between a learning problem and broader tasks.

**Internal validity.** In the evaluation literature, internal validity is about consistency *within* the specified context of the experimental setup [92]. In machine learning evaluations, we use internal validity to refer to validity properties within a learning problem. If these properties are not satisfied, then the experimental measurement itself is invalid. Examples of internal validity problems in machine learning are comparisons to insufficient baselines or overfitting from test set re-use, both of which invalidate claimed improvements over the state-of-the-art on a given learning problem.

**External validity.** External validity is about the ability to extrapolate – to make valid conclusions for contexts outside the experimental parameters [92]. In machine learning, we use external validity

to refer to connections between specific learning problems and the broader tasks they are meant to represent. This goes beyond test set performance on an individual learning problem and is anchored to expectations for performance on one learning problem to transfer to other related learning problems. For instance, external validity issues can arise from limitations of the benchmark dataset or a mismatch in the evaluation metrics of interest.

Internal validity criteria are well known in the field. But despite the seeming simplicity of these failure modes, their recurrence across different areas indicates that machine learning currently has not yet identified nor implemented mechanisms needed for rigorous evaluation. The in-depth study of external validity criteria has only begun recently as more research datasets and concrete applications have become available. Since many popular machine learning benchmarks do not represent real applications but instead are constructed solely for the purpose of comparing learning algorithms, investigating the external validity of these benchmarks is particularly important.

## 3    Internal validity

In this section, we provide examples of recurring *internal validity* issues that arise within the benchmarking paradigm. In particular, we discuss implementation variations, errors in test set construction, overfitting from test set reuse, and comparisons to inadequate baselines.

### 3.1    Implementation variations

Different implementations of the same algorithm or metric should behave as close to identical as possible. Variations in behaviour can cause variations in performance, making comparisons difficult if it is unclear which implementation is being referred to. This can result in situations where multiple implementations of ostensibly the same algorithm are effectively distinct methods. We describe specific cases of implementation variations leading to internal validity failures here, and continue with more examples in Appendix B.1.

*Algorithms*. Ancillary details of an algorithm implementation, often dubbed "tricks", can significantly affect performance. These details are often undocumented in the paper, so subsequent implementations of the algorithm are coded differently. Consider the variation observed by [59] for algorithms in deep reinforcement learning (deep RL): across three implementations of Trusted Region Policy Optimization (TRPO) [134], and three implementations of Deep Deterministic Policy Gradients (DDPG), the best codebase was several factors better than the next best. On OpenAI HalfCheetah-v1 [19], the best TRPO codebase found by [59] achieved an average reward of nearly 2,000 versus 500, and the best DDPG implementation reached a best average reward of 4,500 versus 1,500.

Combining "tricks" employed in various implementations may produce a new, superior algorithm. For example, a collection of different tricks was sufficient in 2018 for a four percentage point top-1 accuracy increase on ImageNet for the ResNet-50 architecture [58], leapfrogging newer and supposedly improved models, like SE-ResNeXt-50. One of the tweaks was first found in a particular implementation of ResNet before adoption by subsequent papers, highlighting that these changes are not broadly documented. Along the same lines, [11] found in 2021 that using an appropriate scaling of the architectural dimensions and image resolution, along with a bag of tricks such as those from [58], can actually outperform the more recent EfficientNets [147].

*Metrics*. Unexpected differences in metric scores caused by implementation variations hinder proper comparisons. In machine translation, the widely-used BLEU score [111] depends on certain parameters which are often unspecified, such as the maximum n-gram length. Further, researchers can silently manipulate the score with changes like adding or removing tokenization, or lowercasing text [115]. Tweaking all these levers in unison results in BLEU score variations of as much as 1.8 BLEU [115] (for context, the gap between the #1 and #2 for one MT dataset as tracked by Papers with Code is 0.14 BLEU [110]). The use of a standardized library such as SACREBLEU [115] to ensure reproducible parameters can help alleviate issues with metric implementations.

*Libraries*. Research code relies on frameworks and libraries to implement common functions. If these libraries aren't coded correctly, evaluation is undermined. Between the Python Image Library (PIL), PyTorch, OpenCV, and TensorFlow, only PIL correctly downsamples a circle without introducing aliasing artifacts [112]. Consequently, implementations of the Frechet Inception Distance (FID) [63], which is used to evaluate generative models, would report different scores for the same models [112].

## 3.2 Errors in test set construction

Even if implementations of algorithms are reliable, flaws in a test set's construction can distort the performance reported on a given learning problem in a few different ways.

*Label Errors*. Several researchers have long articulated concern for the correctness of data labels as an indicator of internal validity [17, 105]. However, it remains unclear how much such errors impact performance measurement, if at all, especially for deep learning [139]. A subset of label errors are due to more conceptually consistent disagreements between annotators [27] or dataset bias [146]; these types of errors are more appropriately construed as external validity issues, and are described further in Section 4.4.

*Label Leakage*. At times, data features accidentally contain direct information about the target variable in a way that makes the learning problem redundant [70]. For instance, a bank account number could be included as a feature to predict the individual has an open account.

*Test set size*. Evaluating a model on a finite-sized test set always leaves uncertainty about the actual performance on the underlying distribution the test set is sampled from. If a test set is too small to detect performance differences between two models, random variation in the test set scores can lead to misinterpreting one method as superior to another [16, 22]. In Appendix B.2 we provide more technical details about appropriate test set sizes.

*Contaminated Data*. Flaws in the dataset construction process may lead to unintentional inclusions of examples that cause problems during evaluation. For example, [8] find that 10% of the images from the CIFAR-100 [76] test set have duplicates in the training set. After deduplication, model performance drops by as much as 14% (relative), demonstrating that the contaminated data leads to overestimation of model performance. Similarly, cross-validation or testing on time-series must be handled with care so as to not include future data in the training set [23]. Examples which are not drawn from the distribution of interest can also distort apparent model performance. Machine translation models perform worse on test sets with more translation artifacts [80]. Models perform up to twice as well on test sets that exclude certain kinds of poor translations as they do on test sets which don't filter these examples out.

## 3.3 Overfitting from test set reuse

When evaluating a model on a test set, we are not interested in performance on the specific test examples, but more generally in performance on similar data. Formally, we hope that the model generalizes to data from the same distribution. The connection between the test set and its corresponding data distribution is only guaranteed if the test set is not reused frequently. This is a core assumption in test set evaluations and is commonly recognized in lecture notes and textbooks [56, 100].

Researchers routinely undermine this assumption by repeatedly reusing popular test sets for model selection, raising concerns about the validity of benchmark results. However, even decade-long test set reuse has surprisingly resulted in little-to-no overfitting on popular benchmarks such as MNIST, CIFAR-10, ImageNet, SQuAD, the Netflix Prize, and more than 100 Kaggle classification competitions [97, 123–125, 128, 163]. While these findings are good news for the benchmark paradigm, they also illustrate that our understanding of common evaluation practices is still limited. An active line of research investigates the question of overfitting from test set reuse, also known as adaptive overfitting [5, 9, 14, 37, 42, 91, 175]. Note that the cited experimental studies of overfitting mostly focus on classification. Regression benchmarks may be more affected by test set reuse.

## 3.4 Comparison to inadequate baselines

Finally, reliably tracking progress on a learning problem requires comparing new methods to existing baselines. In practice, many subtle considerations must be addressed to make proper comparisons. We highlight the biggest recurring themes here; Appendix B.4 contains additional discussion.

### 3.4.1 Implementing and tuning simple methods

Researchers in machine learning often employ newer, more complex methods, such as those using deep neural networks, to solve a given task, without leveraging simpler methods such as linear models or random search. Attention to smaller details and thorough feature engineering can often make a huge difference for these simple baselines:

- In graph learning, logistic regression combined with simple feature engineering provided comparable performance to neural networks while being orders of magnitude faster [67, 162].

- In recommender systems, [25, 125] found that a well-tuned vanilla matrix factorization baseline with some feature engineering outperformed all newer methods, both neural and non-neural, on recommendation results and collaborative filtering tasks.
- In reinforcement learning, where simple linear or RBF policies were able to solve an array of continuous control tasks [118].
- In information retrieval, where a non-neural method from 2004 is superior to all neural approaches developed through 2019 [164].
- In few-shot classification, where a linear layer on top of a supervised classifier's features provides competitive performance on meta-learning benchmarks [151].
- On tabular clinical prediction datasets, where standard logistic regression was found to be on par with deep recurrent models [10].
- And in adversarial robustness, where early-stopping with standard projected gradient descent was found to give performance on par with newer alternatives [127].
- On 3D reconstruction tasks, simple clustering and retrieval in the embedding space outperforms state of the art reconstruction networks [149].

Random search is also frequently overlooked, even though it forms a strong, simple, baseline where applicable. One particularly prominent case is in deep RL, where simple random search, combined with a handful of minor modifications, outperforms many deep RL algorithms on a variety of MuJoCo continuous control tasks [90]. Similarly, for hyperparameter tuning, [79] found that random search combined with early stopping outperformed all existing approaches. And in neural architecture search, [78, 166] found that random search with early stopping and weight sharing found solutions comparable to leading strategies using deep learning. It should be noted that recent NeurIPS competitions found that Bayesian optimization is superior to random search in many settings [155].

### 3.4.2 Controlling for algorithmic details

Implementations of algorithms often contain details to improve performance which are not described in the text. For example, extensively tuning hyperparameters is often key to achieving optimal performance for a proposed method. Unfortunately, baselines are often not tuned as carefully, inflating apparent gains for the proposed method. Ignoring these consequential details leads to misattributions of why one algorithm is better than another, affecting future research directions. For instance, a series of recent papers have attempted to benchmark a variety of deep metric learning algorithms, controlling for aspects such as network architecture, optimizer, image augmentations, hyperparameter compute budget, etc. [41, 101, 129]. After controlling for these factors, the performance difference for the best methods were marginal at best, and the papers concluded that the majority of perceived gains could instead be attributed to newer methods using significantly better backbone architectures (e.g., ResNet50 instead of GoogleNet) and unequal hyperparameter compute budgets. These results very closely mirror results from a variety of other settings, such as deep semi-supervised learning algorithms [108], graph neural networks [36, 140], domain generalization [53], and generative adversarial networks [88]. Inconsistencies in backbone architectures and unequal tuning budgets was a common, recurring failure mode across these papers.

We now highlight a particular example where failing to control for algorithmic details led to a significant misattribution of a method's performance gains. According to [39], deep policy-gradient algorithm Proximal Policy Optimization (PPO) [135] contains several refinements, such as improved initialization methods and reward scaling and clipping, over its predecessor, Trusted Region Policy Optimization (TRPO) [134]. These "code-level optimizations" [39] are a modular addition, and removing these tricks from PPO results in it performing worse than TRPO with them. The improvement from adding the code-level optimizations is larger than switching the underlying algorithm, even though PPO still outperforms TRPO when both (or neither) have the code-level optimizations.

### 3.4.3 Human baselines

Finally, when used for user studies, human baselines are often poorly computed. Based on the context and expertise of the annotator, there are fundamental inconsistencies in human performance, leading to local perceptions of performance that are far from universal. A study measuring human performance on ImageNet using five labellers (three with more training) found errors of 8.1%, 5.3%, 4.3%, 3.8%, and 2.7% [139]. According to the authors of SQuAD 1.1 [121], a natural language question and answering dataset, the human accuracy baseline is likely an underestimate due to using only a single human [120]. Furthermore, assessor ability can impact human baselines — for instance, expert translators agree more and can identify more errors on machine translation benchmarks [152].

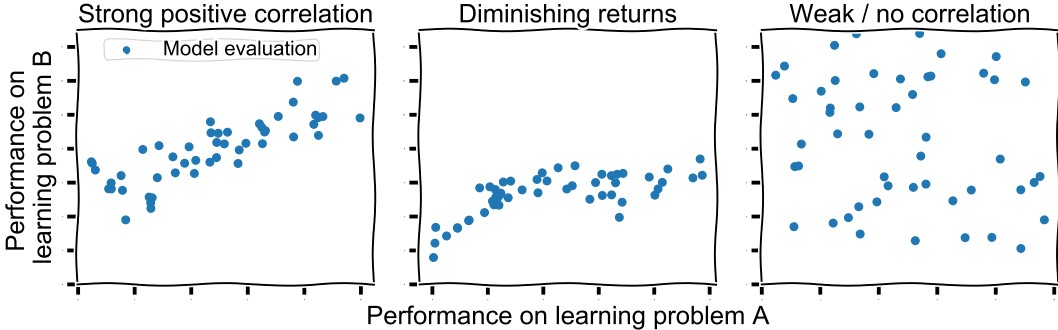

Figure 3: Learning problem transfer can happen to varying extents. Progress on learning problem A may transfer to learning problem B universally (**left**). However, progress may also plateau (**middle**) or there may be no correlation between performance on the two learning problems (**right**).

## 4 External validity

Developing tailored algorithms for specific learning problems is usually not the end goal of machine learning research; rather, the hope is that the ideas and contributions will apply to broader scenarios. How much one expects progress to transfer is a subjective judgment based on factors such as the learning problems involved, the domain knowledge required, and the details of the algorithm itself. We refer to this as *external validity*, as it involves relationships between two or more learning problems. In this section, we first discuss and define two sub-types of transfer that occur within external validity, then provide examples where evaluation issues have arisen.

### 4.1 Types of transfer

*Algorithm transfer.* The claim that a certain algorithm "generalizes well to other problems" is a claim about *algorithm transfer*: the correlation between (i) the relative performance of an algorithm over one or more baselines on one learning problem to (ii) the relative performance of the same algorithm over a one or more baselines on another learning problem. Consider ResNets [57] when they were introduced: adding residual connections (allowing for a deeper net) lead to better performance on ImageNet than VGG [142], a baseline algorithm. On CIFAR-10, ResNets also outperform VGG, an appropriate baseline choice, so we say that ResNets transfer well from ImageNet to CIFAR-10.

*Learning problem transfer.* Now we introduce *learning problem transfer*: the correlation of performance trends over all algorithms for one learning problem with performance trends over all algorithms for another learning problem. Whereas algorithm transfer is about the relative performance of a specific algorithm between learning problems, learning problem transfer asks about the relative progress of algorithms in general between learning problems. For example, as models have improved on the ImageNet benchmark, the same models are used on the CIFAR-10 benchmark, and show continued progress there also. If algorithms never transferred well between learning problems, then progress on one learning problem would never transfer to another. This is visualized in Figure 3 (right), which illustrates low or no correlation between performance on two learning problems. If the correlation weakens over time, this is the "diminishing returns" scenario shown in the middle subplot. And if there is strong positive correlation, then the picture is similar to the first subplot.

Achieving progress in machine learning requires progress on "friendly" learning problems which exhibit strong learning problem transfer; otherwise, researchers would have to start from scratch on every novel learning problem. How can we predict how performance will correlate between two learning problems? There are some common patterns in the literature that allow us to more concretely grapple with learning problem transfer. The community has developed specific out-of-distribution (OOD) test sets for certain problems, such as image corruptions in image classification [60], heuristics-based counterexamples within language inference [94], and a number of "in-the-wild" distribution shifts [6, 61, 62, 74, 124, 159]. Cast in terms of our framework, these OOD benchmarks alter the data distribution of the learning problem, but otherwise remain very close to the original learning problem in the task hierarchy. On the other hand, one may consider transfer of progress between learning problems that are further apart in the task hierarchy, such as from image classification on ImageNet to image segmentation on COCO. In general, as Figure 2 illustrates, the closer two learning problems are in one's conception of the task hierarchy, the greater one may expect positive transfer of progress.

Leaving a more fine-grained discussion of the various of categories of transfer to Appendix A.2, we now explore examples from the literature pointing out failures of learning problem transfer. Since

a learning problem is defined as a dataset plus a metric, a failure in transfer can be attributed to either a misalignment in the datasets or a misalignment in the metrics. Such a misalignment reflects the inconsistencies that arise when boiling down an idealized task into concrete learning problems. Resolving these inconsistencies in either the dataset or the metric may require re-annotating the data or collecting new data; therefore, misalignments are usually baked into the benchmark once the dataset has been constructed and the design choices locked in. All future modeling work on the benchmark inherits the same misalignment problems, underscoring the need for a better understanding of the external validity of commonly used benchmarks.

## 4.2 Metrics misalignment

We use *metric* to mean any algorithm or procedure which, given a model and a dataset, returns a number or score which is interpreted as the performance of the model on that dataset. This definition encompasses not only mathematically defined metrics like accuracy, precision, and recall, but also metrics parameterized by models (Frechet Inception Distance [63], BERT [111], BLEURT [137]), and metrics which involve humans in the loop, like human evaluations of machine translation (Direct Assessment [144], Relative Ranking [51]).

A metric which fails to adequately distinguish between two algorithms that perform differently fails to capture what it means to do well on the learning problem. For example, a good representation learning algorithm should cluster items of the same class together tightly and separate clusters of different classes widely. Papers for representation learning usually report the F1, Recall@K, and Normalized Mutual Information (NMI) metrics. However, all three metrics fail to reward algorithms which have a greater separation between different classes [101]. Even more egregiously, NMI returns higher scores for datasets with more classes, regardless of the algorithm's performance [101].

Researchers may prefer to measure an idealized metric whose use is precluded by practical considerations like money or time, and therefore substitute another metric instead to form a proxy learning problem. For example, many have argued that human evaluation is the 'gold standard' for machine translation [50, 69, 87], but waiting for humans to evaluate translations takes much longer and is much more expensive than computing BLEU [111], an automatic metric. In certain cases, human rankings of translations contradict the BLEU ordering [38, 171].

## 4.3 Comparisons to human performance

Comparing algorithms to humans requires more nuance than any one given learning problem provides. Matching a human baseline on a specific learning problem does not automatically imply human-level performance on other similar similar problems without more evidence. For one, instantiating a task into some learning problem often strips out context which meaningfully affects evaluation. In translation, for example, the work of human translators tends to be evaluated as a complete text, whereas machine translation competitions compare hypothesis sentences to reference sentences, meaning that erroneous translations which are apparent only in context are missed [152].

Further, claims to "super-human" performance on a given learning problem is related to but does not always translate to "human-like" reasoning or ability [44] – for instance, contemporaneous models suffer performance drops with only small changes of the learning problem that don't affect humans as badly (e.g. models [64] on CIFAR-10 [76]). Claimed improvements by themselves are thus only applicable to the given learning problem, and aren't sufficient to prove machine superiority on the broader task or application.

## 4.4 Dataset misalignment

Specific decisions made about data collection and curation are increasingly acknowledged as highly consequential to model outcomes [113, 132]. Any failure to transfer from one learning problem to another learning problem or broader task is often tied to the data choices involved. Because of the cost and effort involved in annotation and data collection, these decisions can have a broader impact than failures contained to a single modeling paper. In the next two subsections, we explore how specific choices in dataset curation can hinder an algorithm's ability to transfer. Refer to Appendix B.5 for additional discussion and examples.

### 4.4.1 Reliance on simple, inappropriate heuristics

We found several examples where gaps in the data collection process lead to models performing well on a given learning problem by relying on data quirks which do not characterize the overall task. For instance, [107] discovered that sub-par clinical performance of X-ray image classification models

was in part due to an unintended correlated variable in the training data: classifiers trained to predict whether an X-ray image presented a collapsed lung were failing disproportionately on new positive diagnoses. It was discovered that a majority of the positive training images actually contained visible chest drains, a treatment for the condition. Thus, models achieved a high accuracy on the learning problem by identifying whether a chest drain was present, but completely sidestepped the original purpose of the task. After removing the spurious feature, by filtering out chest drain images, model performance dropped significantly, by over 20% on clinically relevant subsets of the data.

More examples of models exploiting simple dataset-level heuristics abound. The authors of [49] found that on the Visual Question-Answering dataset [4], models could exploit strong label imbalance on certain questions. For example, for a question beginning with "Do you see a...", a model always outputting "yes" – without considering the rest of the question or the actual image – can achieve an accuracy of 87%; correcting this imbalance in the test set led to accuracy drops of up to 12% among yes/no questions for these models. Similarly, models trained on part of a reading comprehension task (either questions only or passages only) achieve a surprisingly high accuracy [71]. In another case, a benchmark for equation verification (determining if an equation statement is true or false) was revealed to include false axioms and data generation rules that biased the results towards an overwhelming number of false statements [28].

Landmark studies found that language models regularly exploit such "spurious patterns" across a wide range of NLP tasks [46, 72]. On the MNLI natural language inference benchmark, the presence of a negation operator (e.g. "not", "no", etc.) dictates the label probability to a greater degree than the actual input prompts [94]. Similarly, the authors of [104] find that BERT models trained on comprehension datasets (e.g. ARCT [54]) exploit the presence of negation operators, and removing such cues drops the model to random chance accuracy. These correlates were discovered by using humans to augment the training data to be consistent with counterfactual labels. When evaluated on these counterfactual subsets, model performance drops by as much as 30% in multiple cases.

### 4.4.2 Sensitivity to real-world distribution shift

There are also many cases where an algorithm is expected to perform in a broader variety of scenarios than it is trained on. In such cases, the inability to transfer is not caused by exploiting specific obvious heuristics as much as it is caused by a failure to extrapolate to different real-world data distributions. For example, most models trained on ImageNet were found to experience a considerable drop in accuracy when exposed to images that contained a larger amount of natural variation, such as changes in pose, lighting, object composition, etc. [148]. Similarly, models trained for the original SQuAD dataset performed poorly when evaluated on data collected from different source domains, such as Amazon crowd reviews and Reddit posts [97].

In the medical domain, models developed in one institution for diagnosing pneumonia in radiographs or classifying pathology tissue slides may not translate to other hospitals for practical reasons such as differences in equipment and patient populations [74, 167]. Similarly, [73] find in a learning problem transfer analysis from ImageNet to chest X-ray classification on CheXpert [68] that, while ImageNet pre-training helps models achieve higher performance on CheXpert, models with higher ImageNet accuracy are not likely to provide higher CheXpert performance.

### 4.4.3 Dataset Bias & Disagreement

At times, the misalignment perceived between the learning problems is the result of various forms of data bias [146]. Some data sources can omit or under-represent certain sub-populations and as a result, evaluation measurements will disguise failures for these under-represented population subgroups [119]. For example, facial recognition benchmarks drastically under-represent darker and female faces [96], making it difficult to perceive when models fail to perform acceptably for this subgroup [7, 20]. Furthermore, inappropriate stereotyped associations can be perpetuated by the systematic use of offensive, incorrect or exclusionary labels for certain mistreated subgroups [116, 145]. At times, societal discrimination can also lead to false labels being more common in one group than another [99]. Discrepancies between learning problem datasets may also arise from inherent contextual differences - data sourced from differing geographies or cultural context [29, 138], in addition to annotators with inherently differing viewpoints regarding ground truth [27, 48].

### 4.5 Evaluation quantification

The aforementioned examples of metric and dataset mislignment suggest that reliably measuring progress in machine learning requires evaluating on multiple learning problems associated with a

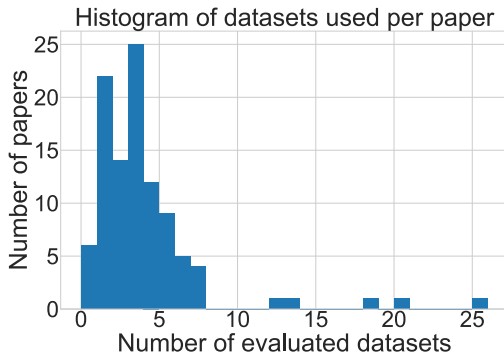 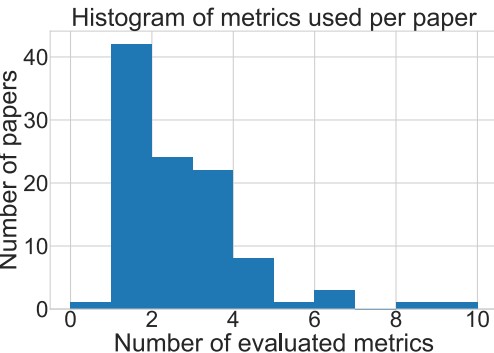

Figure 4: A histogram of the number of datasets used for evaluation by each paper in our sample pool (**left**), and a similar histogram for the number of metrics (**right**). Most of the papers (>65%) evaluate on 3 datasets or fewer, and a similar fraction (>65%) evaluate on 2 or fewer metrics.

particular task. If a proposed method provides gains in a variety of different contexts, one can be more confident in the performance on future learning problems instantiating the task.

To better understand community practices around benchmarking and provide some context around our analysis and framework, we annotated a random sample of machine learning benchmarking papers with the number of distinct datasets and the number of distinct metrics each paper used for evaluation. Concretely, we randomly sampled 140 papers from the past five years (2016–2021) of NeurIPS, ICML, EMNLP, and CVPR, and filtered out papers which were not applicable to the benchmarking paradigm (37 papers). The results of our analysis for the remaining 103 papers are presented in Figure 4. On average, papers evaluated on an average of 4.1 datasets and 2.2 metrics. Overall, most of the papers in our sample (>65%) evaluate on 3 datasets or fewer, and a similar fraction (>65%) evaluate on 2 metrics or fewer. Although we cannot recommend a "correct" number of learning problems to evaluate on, as this is a domain-specific consideration based on the task and specific learning problems, our data provides evidence that many papers evaluate on a small number of datasets and metrics, which indicates that studying alignment between these learning problems can be a helpful guide for future research. We provide more detail about our paper collection and annotation procedure, as well as confidence intervals for our mean estimates, in Appendix C.

## 5   Broad critiques of benchmarks & competitive testing

Researchers have described several limitations to the benchmarking paradigm in machine learning. Most obviously, the use of benchmarks to assess progress in the field creates a competitive testing dynamic that emphasizes outcomes rather than proper scientific inquiry [66]. The absence of community norms like reproducibility guidance [34, 114], documentation standards [98] or statistical significance testing [16] makes relying on outcomes-based approaches to evaluate progress even more questionable [13]. Behavior-based alternatives to the benchmarking paradigm, such as test suites [1, 126, 170], for example, can re-orient ML evaluation away from its current focus on the competitive determination of "state of the art", and more towards an exploratory and descriptive probing of model capabilities [65, 106, 143, 161, 169]. Furthermore, the learning problems we embody as benchmarks go a long way in focusing community attention on a set of specific applications and tasks, not all of which are ideal or value-aligned. For instance, the lack of consideration for other aspects of performance in ML evaluation, such as model efficiency, privacy or fairness, plays a big role in disincentivizing researchers from paying attention to such issues [40, 136].

## 6   Conclusion

The benchmarking paradigm has served as a valuable guide for progress in the past. However, the next phase of machine learning innovation and deployment will require more sophisticated evaluation practices than comparing one-dimensional performance numbers on a single test set. We hope that our taxonomy offers a starting point for both experimental and theoretical research in this area, and that the field will invest in a more robust understanding of the evaluation practices that inform our shared perception of progress.

