# Appendix

Our paper involves a meta-review of survey papers commenting on different evaluation failures in machine learning across various sub-disciplines. Tables 2 and 3 provide an overview of the papers in our survey. As is evident from the distribution of papers, internal validity issues are more commonly discussed in machine learning. However, recently there has also been increasing interest in external validity, particularly in natural language processing and computer vision.

We do not claim that our survey is exhaustive – it was compiled through extensively crowd-sourced recommendations from peers, search engine queries, following citation graphs, and though our own awareness of the field and expertise. As we read through the surveyed papers, we noted down the main evaluation failures each paper brought up. We then came up with the failure mode organization in a bottom-up processs, through extensive discussion about the papers, making sure that each paper fit within the proposed taxonomy. We also paid attention to making sure that our categorization was formally sound by covering all the cases from the initial research idea to progress on a broad variety of real-world applications, as outline in Figure 1. The survey is meant to serve as guidance and evidence of a broader emerging discussion in machine learning.

In Appendix A, we revisit key concepts from the paper, elaborating on our definitions of internal and external validity as well as detailed descriptions of transfer types and categories. Appendix B extends the discussion of individual failure modes, providing examples that we were not able to include in the main text and recommendations for avoiding the common pitfalls in each failure mode. Appendix D contains the aforementioned tables indexing the included survey papers. In Appendix E, we have included a very short summary of each paper in our meta-review.

# A  Additional discussion of main concepts

We now provide further details for the concepts presented in the main paper, grounding our points by formalizing certain terms. For simplicity, the concepts are stated in the context of supervised learning, extending the concepts to other learning setups is straightforward.

## A.1  Learning problems and internal validity

The core element of study in machine learning is a *learning problem $L$*, which, as mentioned in Section 2.1, consists of a *dataset $S$* and an evaluation metric $\alpha$, e.g., accuracy. The associated definitions are well established in the literature. The dataset $S$ is split into two parts: a training set $S_{\text{train}}$ and a test set $S_{\text{test}}$, with both parts containing examples $(x, y)$ comprised of features $x \in \mathcal{X}$ and labels $y \in \mathcal{Y}$. The feature space $\mathcal{X}$ is often a real vector space (image pixels, demographic information, etc.) and the label space $\mathcal{Y}$ is often a discrete set of classes or the real numbers.

The goal of machine learning is to find an algorithm $A : (\mathcal{X} \times \mathcal{Y})^{n_{\text{train}}} \to (\mathcal{X} \to \mathcal{Y})$ that maps the $n_{\text{train}} = |S_{\text{train}}|$ training examples to a model $f : \mathcal{X} \to \mathcal{Y}$ that generalizes beyond the training examples. Here we use "algorithm" broadly to incorporate all aspects such as the model architecture, optimization method, and hyperparameter choices. To evaluate the generalization capability of a trained model $f$, we compute its performance $\hat{\alpha}(f, S_{\text{test}})$ on the $n_{\text{test}} = |S_{\text{test}}|$ examples in the held-out test set (we overload earlier notation for the evaluation metric $\alpha : \mathcal{Y} \times \mathcal{Y} \to \mathbb{R}$).

The test set is not the final goal of building the model (after all, the test set labels are already known). Instead, the test set serves as a proxy for performance on similar future data. This idea is usually formalized by positing that the test examples are drawn from a distribution $D_{\text{test}}$. The test set performance $\hat{\alpha}(f, S_{\text{test}})$ is the finite-sample approximation of the population performance $\alpha(f, D_{\text{test}})$, which is the ultimate quantity of interest.

**Internal validity.**  In the evaluation literature, internal validity is about consistency *within* the specified context of the experimental setup [92]. In the case of machine learning, we use "internal validity" to refer to validity properties within the context of an individual learning problem. The core question in a benchmark evaluation on a learning problem is whether a claimed performance improvement of a new model $f$ compared to a prior baseline $f_{\text{baseline}}$ really exists, i.e., if the comparison between $\alpha(f, D_{\text{test}})$ and $\alpha(f_{\text{baseline}}, D_{\text{test}})$ is valid. This comparison can be affected by internal

validity problems such as implementation details (Section 3.1), insufficient baselines (Section 3.4), overfitting from test set re-use (Section 3.3), and dataset construction issues (Section 3.2).

## A.2 External validity – transfer definitions and categories

When a machine learning paper reports algorithmic improvements on a specific learning problem, the learning problem itself is actually rarely the focus of the research effort. Instead, the paper implicitly or explicitly claims to improve performance on a broader *task*. This involves a generalization of the improvement observed on one learning problem to other contexts, other environments, and other related tasks.

As a concrete example, most papers evaluating on ImageNet do not frame their contributions as progress on classifying Flickr images (filtered in a special way) into one of 1,000 classes (which include more than 100 dog breeds). Despite its importance, the ImageNet (ILSVRC-2012) learning problem is a toy task. So instead of focusing on the specific learning problem, a paper frames its experiments as an instantiation of a more general task such as "image classification". The ambition of a paper is that its algorithmic contribution applies to a wide range of problems under the umbrella of this more general task. But what this more general task exactly is, how broadly the proposed method applies, and how the learning problem relates to the task is rarely if ever made explicit.

The relationship between learning problems and tasks underlies most evaluations in machine learning. Since this relationship is rarely explicitly studied, the subtle connection leads to various failures in machine learning evaluations. In the evaluation literature, such concerns fall under the umbrella of "external validity", which is about the ability to extrapolate – to make valid conclusions for contexts outside the experimental parameters (i.e., outside the specific learning problem). In this paper, we describe the external validity issues of dataset construction (Section 4.4), metrics (Section 4.2) and comparisons to human performance (Section 4.3). Next, we introduce a formalization and more detailed description of the transfer concepts presented in Section 4.1 of the main text.

Table 1: Overview of Transfer Categorization

|  | Test-time transfer | Train-time transfer | General transfer |
|---|---|---|---|
| **Algorithm contribution transfer** | Does a specific algorithmic improvement over a baseline benchmarked on $L_1$ hold when benchmarked on $L_2$? | Does a specific algorithmic improvement over a baseline benchmarked on $L_1$ hold when re-trained and benchmarked on $L_2$? | Does a specific algorithmic improvement over a baseline benchmarked on $L_1$ hold when re-formatted, re-trained and benchmarked on $L_2$? |
| **Learning problem transfer** | Do model improvements benchmarked on $L_1$ generally correlate with improved performance when tested on $L_2$? | Do model improvements benchmarked on $L_1$ generally correlate with improved performance when re-trained on $L_2$? | Do model improvements benchmarked on $L_1$ generally correlate with improved performance when re-formatted and re-trained on a more conceptually distinct $L_2$? |

### A.2.1 Performance transfer

Performance transfer is the extent to which performance on one learning problem leads to performance on another learning problem. More formally, let $D_1$ be the test distribution of the first learning problem (associated metric $\alpha_1$), and let $D_2$ and $\alpha_2$ be the test distribution and performance metric of the second learning problem. Finally, let $f_1$ and $f_2$ be the models produced by the same learning algorithm when applied to the two learning problems, respectively. Then the question of transfer is what $\alpha_1(f_1, D_1)$ implies for $\alpha_2(f_2, D_2)$.

Often the absolute performance numbers on two different learning problems are not directly comparable. One reason is that two learning problems can have different difficulty (e.g., varying label noise or training set sizes), which means that the same learning algorithm yields models that achieve different performance values $\alpha_i$. Another reason can be that the two metrics $\alpha_1$ and $\alpha_2$ may quantify different performance aspects (e.g., classification error or a regression loss). Hence understanding the transfer between two learning problems often involves the performance *improvements* achieved by a proposed algorithmic intervention.

More precisely, let $I$ be the intervention to the learning algorithm, let $f_1$ be the baseline model (learned without intervention), and let $f_1^I$ be the model learned with intervention (similarly for $f_2$ and $f_2^I$). Then $\Delta_i^I = \alpha_i(f_i^I, D_i) - \alpha_i(f_i, D_i)$ is the improvement on learning problem $i$ achieved by intervention $I$. If the intervention $I$ transfers from learning problem 1 to learning problem 2, $\Delta_1^I$ and $\Delta_2^I$ should be consistent (up to scaling to accommodate different metrics $\alpha_i$).

More broadly, claims of performance for an entire *task* $T$ need to transfer to all learning problems within the scope of the claimed task. Specifically, let $L_1, L_2, \ldots$ be learning problem instantiating the same task $T$, and let $\Delta_i^I$ be the performance improvements achieved by intervention $I$ on learning problem $i$ as before. Then intervention $I$ improves on the entire task if and only if the $\Delta_i^I$ are consistent. If one of the $\Delta_i^I$ behaves differently, either learning problem $i$ is not actually an instantiation of the task, or the intervention $I$ is less effective on this learning problem, adding an important caveat to the claimed broad improvement on the entire task.

### A.2.2 Types of Performance Transfer

We distinguish two types of transfer – *learning problem transfer* and *algorithmic transfer*.

For *learning problem transfer*, we track how closely performance improvements on one learning problem correlate with performance improvements on a different learning problem *for all algorithms*. This indicates to us how much progress on one learning problem signifies progress on another, and thus progress on a broader set of representations of a task. Here we expect the performance improvement $\Delta_i^I$ of any intervention $I$ on a learning problem $i$ instantiating task $T$ to translate to a performance improvement when $T$ is represented by another learning problem. This means improvements on one learning problem should correlate with improvements on all other learning problems for the task, for all algorithmic interventions.

For *algorithmic transfer*, we expect a *specific* algorithmic intervention $I$ to yield a performance improvement $\Delta_i^I$ over the baseline on any learning problem for a given task $T$. So if the intervention $I$ leads to progress on this task $T$, $\Delta_i^I$ should be consistent with $\Delta_j^I$ for any pair of learning problems $i$ and $j$ instantiating the task $T$, given the assumption that the learning problems adequately represent the task $T$.

### A.2.3 Common Transfer Categories

Transfer is always between two learning problems. For both types of transfer, our expectation for how much transfer we expect between learning problems is dependent on their similarity. In other words, the more similar two learning problems, the more we expect performance improvement to transfer between them, and the more we expect performance to correlate on them.

As tasks are semantically defined, it is difficult to define with certainty how similar two learning problems are – all we can say is that similarity lies along some spectrum, where we can range from nearly identical learning problems to learning problems with so many differences they could be perceived as representing conceptually separate sub-tasks of task $T$. We attempt to demonstrate the range of this spectrum, by introducing three familiar examples along the spectrum: Transfer categories A, B, C - with Transfer A occurring between more similar learning problems and thus being more expected but Transfer C, between less similar problems, being less expected. We thus define these three common transfer categories along the spectrum of transfer expectations, tied to the similarity of learning problems. These scenarios are described with examples, below.

**Definition A.1. Transfer A: Changing the learning problem at test time.**

We often deploy systems in a slightly different context from that in which they are developed. As a result, real-world test environments often have small changes to the dataset or metrics, and thus

we make small changes to the learning problem at test time. Regardless of these small changes, we expect to end up with the same task, and thus a comparable performance improvement on this task.

As mentioned earlier, a *learning problem* $L$, consists of a *dataset* $S$ and an evaluation metric $\alpha$. The difference between learning problems in Transfer A is due to updates to a test set $S_{\text{test}}$, as a change in the distribution of features at test time (the label space $\mathcal{Y}$ often remains constant). Transfer A could also involve the use of an alternative metric $\alpha$ at test time. For example, a CNN model trained on ImageNet[130] should have a comparable improvement over baseline performance on a test set from ImageNetV2 [148], with the same labels but a different data distribution. In the same way, a CNN model trained on multi-label accuracy should have a comparable improvement over baseline performance when evaluated for performance on single label accuracy.

**Definition A.2. Transfer B: Changing the learning problem at training time.**

Further along the spectrum is a situation involving a less similar set of learning problems - usually involving a change between the learning problems so notable that the model needs to be re-trained in order for performance to be evaluated on this new learning problem.

Transfer B often involves learning problems in which the dataset $S$ and an evaluation metric $\alpha$ are different - including both $S_{\text{test}}$, and $S_{\text{train}}$. This could involve updates to both features $x \in \mathcal{X}$ and labels $y \in \mathcal{Y}$. Transfer B could also involve the use of an alternative metric $\alpha$ that requires model re-training. For example, training a CNN model on ImageNet [130] and evaluating if algorithmic improvements are a consistent improvement if the model is retrained to be evaluated on CIFAR-10 [64]. One could also re-train the model to compare performance on learning problems of differing metrics.

**Definition A.3. Transfer C: Changing the task.**

A pair of learning problems are most different when they diverge to the point of being able to be conceptualized as semantically distinct tasks. For example, if $L_1$ is top-1 accuracy on ImageNet[130] and $L_2$ is average precision on COCO [82], then although both learning problems are different enough to be conceived as separate tasks (i.e., "image classification" and "object detection", respectfully), they can also both be seen as learning problems instantiating the task $T$ of image recognition.

Similarly, there is an expected performance transfer across a suite of Natural Language understanding sub-tasks in GLUE [158]. Although the benchmark is composed of various sub-tasks (eg. Question-Answering and Inference learning problems),there is an expectation for models optimized for natural language understanding to be able to perform well on each included learning problem.

# B    Additional discussion & recommendations for each failure mode

In this section, we include additional examples and provide recommendations for addressing each described failure mode in the main text.

We elaborate on the discussion on implementation variations (Section 3.1), baselines (Section 3.4), overfitting (Section 3.3), test set contruction issues (Section 3.2), metrics misalignment (Section 4.2), dataset misalignment (Section 4.4), and human comparisons (Section 4.3).

## B.1    Implementation variations

We discuss here some additional cases of implementation variations affecting evaluation.

The line between a bug and coding something valid but not what was intended is thin. The implementation of the Adam optimization algorithm in `PyTorch` once incorrectly scaled a certain epsilon factor, resulting in a minor difference in behavior [122]. More seriously, many libraries implemented Adam with $L_2$ regularization in lieu of weight decay - an equivalence for standard stochastic gradient descent, but not for adaptive gradient algorithms, of which Adam is one [86].

Common libraries are used across codebases, and silent changes may affect evaluation without widespread realization. For example, in November of 2018, maintainers of `mosesdecoder`, a widely-used text tokenizer for machine translation, changed its behavior in a certain edge case. Specifically, `mosesdecoder` began tokenizing the final period separately from the final word: "I laugh." became ["I", "laugh", "."] instead of ["I", "laugh."] previously. This increases the number of tokens per sentence, affecting how scores were computed by most automatic metrics [115], such as the BLEU

## B.3 Overfitting from test set re-use

Multiple papers have found that overfitting from test set re-use occurs surprisingly rarely, at least in classification problems [97, 123, 124, 128, 163]. So at a high level, our recommendation here is to take other validity issues at least as seriously as overfitting, even if overfitting has traditionally received the main attention when discussing evaluation failures in machine learning [56, 100].

Beyond this high-level point, we offer two recommendations:

*Recommendations.*

- Continue to use validation sets for frequent model selection in addition to a test set that is accessed more rarely. While this classical practice usually offers little advantages, it also comes at little cost in the development workflow. As long as the principles behind test set adaptivity are not understood better, it is safer to use separate validation and test sets.
- Pay special attention when working on regression problems. As mentioned before, the absence of overfitting from test set re-use is currently mainly investigated for classification problems. Anecdotal evidence suggests that overfitting may be more widespread in regression problems, so separate validation and test sets are more important in this domain.

## B.4 Comparisons to inadequate baselines

As an additional example of a comparison to inadequate baselines, small differences in training between BERT models can vary performance substantially, by up to 7% [33]. In fact, by just retraining BERT over multiple random seeds, [33] were able to outperform more recent methods such as XLNET [165] or RoBERTa [84].

*Recommendations.*

- *Simple baselines.* First, it is important to benchmark against well-tuned simple, classical methods, such as random search, linear or logistic regression, boosted decision trees, etc. The inclusion of smaller details can often make the difference with these methods, so care must be taken that enough thought and effort has gone into feature engineering and tuning hyper-parameters.
- *Control for variations.* Second, since seemingly trivial algorithmic details may have an outsize impact on final performance, baselines comparisons should be made on an equal footing, where only the only source of variation is the algorithmic contribution. For example, network backbone architectures should remain fixed when comparing different downstream algorithms such as metric learning or generative adversarial networks (unless, of course, the architecture is the main contribution).
- *Ablate algorithmic details.* Performing thorough ablation studies for the inclusion or exclusion of each algorithmic detail should also help elucidate the finer-grained differences.
- *Equal compute budgets.* Additionally, compute budgets for hyper-parameter tuning and adjustment should also be set fixed across all the methods compared to.
- *Proper human evaluations.* Since humans come with various backgrounds and expertise, human evaluators should represent the desired audience for assessment, and the details of their expertise should be explicitly reported. The conditions for the human assessment need to be as identical as possible for each annotator, and there need to be a large enough number of annotators such that individual variances are averaged out.

## B.5 Dataset misalignment

In this section, we discuss a few additional notable examples of dataset misalignment.

In a similar vein to [107], a study on machine learning for COVID-19 detection through radiographs found that most models exploit spurious correlates that do not hold when deployed in different environments [30]. For example, swapping laterality markers on an image (which indicates the right side from the left side of the patient) with those more common in COVID-19 positive images led the model to predict an increase in odds of COVID-19 for the patient. Together, these spurious associations degraded the model's performance from 99.5% to to 70% in certain situations.

score [111]. Due to this change, someone using `mosesdecoder` before November 2018 to evaluate a machine translation model would compute a slightly different BLEU score than someone evaluating the same model after November 2018. Similarly, any changes to a deep learning framework such as `TensorFlow` or `PyTorch` affect all code using that framework.

*Recommendations.*

- *Provide code.* Authors should provide the code used to run the experiments, or at least code which reproduces the results identically, so that implementation details can at least be discovered by others if they turn out to be important.
- *Specify baseline versions.* Authors should especially take care to link to the code used to run baselines (if from an external code base).
- *Disclose hyperparameters.* Authors should report all hyperparameters and experimental settings, such as environmental parameters.
- *Pin and specify dependencies.* Authors should specify versions of dependencies, such as by pinning dependencies to specific version numbers and providing this information in a 'requirements.txt'.
- *Verify implementations.* Authors may wish to compare the behavior of important dependencies, such as metric implementations, by comparing against multiple variations.

## B.2 Dataset errors

### B.2.1 Test set size

As mentioned in the main text, too small test sets can pose an internal validity issue in the benchmarking paradigm. When a test set is too small, the measured improvement of a new method over a baseline may be a result of sampling randomness, not a true improvement on the underlying distribution. On future data (e.g., another test set from the same distribution), the baseline may then perform as well or even better than the new method, invalidating the claimed performance improvement.

Statistics offers a wide range of tools for estimating quantities from a finite number of samples while taking sampling uncertainty into account. A basic and widely applicable tool are *confidence intervals*. Instead of reporting only a single number (the test set performance), a confidence interval describes a range of possible values that are compatible with the available data. As the sample size increases, statistical uncertainty decreases and the size of the confidence interval shrinks.

In order to incorporate the confidence interval into the performance comparison (instead of only the point estimate, e.g., average test set performance), authors should check whether the baseline performance falls into the confidence interval of the new proposed method. If this is the case, the test set is too small to rule out a spurious improvement that arose only from random chance, and the authors should highlight this in the description of their experimental results. Whether a test set is too small depends on multiple factors such as the performance improvement (effect size), the performance metric, and the statistical significance level (or coverage of the confidence interval). A detailed discussion of confidence intervals goes beyond the scope of this paper. We refer the reader to [160] for a concise definition of confidence intervals and a discussion of common pitfalls. Statistical validity in the sciences and best practices are an ongoing conversation, e.g. see [2].

### B.2.2 Recommendations

- *Cross-reference label annotations and assess annotator agreement.* Ideally labels are tolerably consistent across annotators.
- *Verify label correctness.* Confirm that labels are independently verifiable as correct by a domain expert or informed annotator, both reviewing random samples from the dataset. Allow for third party data audits by making dataset accessible.
- *Optimize data test set size.* Calculate what it means for the dataset to have a size that is statistically significant in reporting results. Aim to have a test set size at least 5-10 % the size of the dataset on which a model is trained.
- *Check for contamination.* Visually review a sub-sample from the test and train datasets and confirm that there are no overlapping or overly similar examples included in the final evaluation.

the model to predict an increase in odds of COVID-19 for the patient. Together, these spurious associations degraded the model's performance from 99.5% to to 70% in certain situations.

A comprehensive study of CNN architectures found that transfer performance of architectures from ImageNet to other image classification datasets varied wildly [154]; however, a similar study focusing on transfer performance of pretrained ImageNet classifiers found a very strong correlation in accuracy to downstream datasets [124].

*Recommendations.* A concrete list of recommendations is hard to provide for cases of dataset misalignment, since the test-time distribution is very often unknown at training time.

- *Evaluate on multiple learning problems.* The best way to provide confidence that a given algorithm or model adequately demonstrates the capabilities needed to solve a given task is to evaluate the algorithm on a number of different learning problems and datasets that are all related to a particular task. If the algorithm is shown to have consistent performance across a number of different scenarios, then one may have greater confidence in the test-time performance at deployment. Of course, this does not rule out the possibility that performance may still significantly degrade due to some unexpected test-time distribution change; thus, all results must still be evaluated with caution.

- *Evaluate in context.* One can set up pilot evaluations to assess model performance in a scenario closely resembling the deployment context, to get a direct measurement of the real-world behaviour of the model.

- *Leverage domain expertise.* Consult domain expertise in the design and development of the learning problem, ensuring that the instantiated learning problem is a valid abstraction or representation of the essential aspects of the broader task.

### B.6 Metrics misalignment

*Recommendations.*

- *On task-specific metrics.* Choosing an appropriate metric is part of designing a suitable learning problem. Metrics need to provide meaningful information about a model's performance. Sometimes a standard metric, like accuracy or F1, is sufficient. In other cases, a task-specific or even learning problem-specific metric may be required.

- *Validate proxy metrics.* When the metric selected is a proxy for another, more meaningful metric, the suitability of the proxy metric must be using data from the relevant distribution.

### B.7 Human comparisons

*Recommendations.*

- *Scoped claims.* "Superhuman" performance can only really be claimed with respect to a specific learning problem. Obtaining human-level performance on one learning problem does not necessarily translate to the model having human-level ability on the broader task.

- *Consistent evaluation settings.* The evaluation context for human assessment of the model should be consistent with the context in which humans typically assess human performance on the same task.

## C  Details on paper survey

We provide further detail about our methodology for the paper survey mentioned in Section 4.5. We randomly sampled 140 papers from the past five years (2016–2021, where available) of NeurIPS, ICML, EMNLP, and CVPR; after filtering out papers that did not fit into the benchmarking paradigm (e.g. purely theory papers, some forms of analysis papers, those without experiments, etc.), we were left with 103 papers that formed the basis for our analysis. For each of these 103 papers, we noted down the number of distinct metrics and the number of distinct datasets used for evaluation, excluding any one-off toy synthetic datasets such as random gaussians or extremely simple gridworlds.

The results of our analysis are presented in Figure 4. Note that one paper evaluated on 57 Atari datasets; we omitted this datapoint during plotting Figure 4 (left) but included it the following analysis.

On average, papers evaluate on around four datasets; the mean of our sample is at 4.10, and the 95% percentile bootstrap confidence interval for the mean from 10,000 resamples is (3.04, 5.47). Papers tended to evaluate on fewer metrics, with the average being around two metrics; the mean of our sample is at 2.21, and the 95% percentile bootstrap confidence interval for the mean from 10,000 resamples is (1.93, 2.51). Overall, most of the papers in our sample (>65%) evaluate on 3 datasets or fewer, and a similar fraction (>65%) evaluate on 2 metrics or fewer.

## D  Survey summary table

Table 2: Summary of Analyzed Papers – Internal Validity Issues

| Failure Modes | Implementation | Baselines | Data (Internal) | Adaptive Overfitting |
|---|---|---|---|---|
| **Natural Language Processing** | [103, 172] | [33, 95, 115, 133] | [17, 22, 80, 93, 171, 174] | [127] |
| **Computer vision** | [112] | [32, 149, 151] | [105] | [123, 124] |
| **Generative models** | N/A | [88] | N/A | N/A |
| **Optimization** | [86] | [79] | N/A | N/A |
| **Meta-learning** | N/A | [78, 166] | C | N/A |
| **Metric learning** | N/A | [41, 101, 129] | N/A | N/A |
| **Learning on graphs** | N/A | [36, 67, 140, 162] | N/A | N/A |
| **Tabular data** | N/A | [10, 43] | N/A | [56, 100] |
| **Reinforcement Learning** | [39, 168] | [3, 59, 90, 118] | N/A | N/A |
| **Information Retrieval** | N/A | [164] | N/A | N/A |
| **Recommender Systems** | N/A | [25, 125] | N/A | N/A |
| **Semi-supervised / Unsupervised** | N/A | [108] | N/A | N/A |
| **General / Other** | [16] | [53, 141] | [28, 113, 132] | [14, 128] |

Table 3: Summary of Analyzed Papers - External Validity Issues

| Failure Modes | Metrics | Data (External) | Human Performance | Critiques |
|---|---|---|---|---|
| **Natural Language Processing** | [15, 21, 38, 40, 45, 51, 52, 109, 126] | [17, 46, 47, 71, 72, 83, 94, 97, 104] | [85, 152] | [40, 126] |
| **Computer vision** | [12] | [30, 73, 75, 107, 117, 148, 154, 167] | [139] | N/A |
| **Generative models** | [173] | N/A | N/A | N/A |
| **Optimization** | N/A | N/A | N/A | N/A |
| **Meta-learning** | N/A | N/A | N/A | N/A |
| **Metric learning** | N/A | N/A | N/A | N/A |
| **Learning on graphs** | N/A | N/A | N/A | N/A |
| **Tabular data** | N/A | N/A | N/A | N/A |
| **Reinforcement Learning** | N/A | [3] | N/A | N/A |
| **Information Retrieval** | N/A | N/A | N/A | N/A |
| **Recommender Systems** | N/A | N/A | N/A | N/A |
| **Semi-supervised / Unsupervised** | N/A | N/A | N/A | N/A |
| **General / Other** | N/A | [26, 28, 49, 113, 150, 156] | [44] | [13] |

# E List of analysis papers and short summaries

**Legend of Failure Modes**

    I Implementation Variations

    B Baseline Issues

  DI Data, Internal validity issues (ie. methodological errors, etc.)

   O Adaptive Overfitting

  DE Data, External validity issues (ie. spurious correlations, data misalignment, etc.)

   M Metrics misalignmennt

   H Comparison to Human Performance

   G General critiques of benchmarking

## E.1 NLP (Translation, Question answering, Natural language Inference)

DE *Can Small and Synthetic Benchmarks Drive Modeling Innovation? A Retrospective Study of Question Answering Modeling Approaches [83]* Explores whether synthetic benchmarks could have driven architectural modeling progress in natural language (instead of SQuAD) and finds agreement between the two types of benchmarks in multiple cases.

H *Putting human assessments of machine translation systems in order [85]*. Authors identify the unawknowledged design decisions that bias the assessment of human annotators that use the relative ranking method to evaluate model performance on 25 translation tasks from the annual Workshop on Machine Translation (WMT) in 2010 and 2011. In particular, the order in which candidate translations are presented is shown to bias human judgement and thus evaluation outcomes.

H *Attaining the Unattainable? Reassessing Claims of Human Parity in Neural Machine Translation [152]* A prior claim that a Chinese to English machine translation system achieved human parity falls through when translationese is removed from the picture. Further, expert annotators, in this case, professional translators, are better able to tell between machine and human translations.

I *Do Transformer Modifications Transfer Across Implementations and Applications? [103]* The authors find that most proposed modifications to the transformer architecture do not significantly improve performance across a variety of benchmarks. They suggest this is because modifications are specific to implementations and applications, and fail to transfer beyond their original niche.

DE, O *The Effect of Natural Distribution Shift on Question Answering Models [97]* Explores a variety of naturally occuring distribution shifts for language models, such as collecting data from various online source domains, and finds these changes in the distribution can have a large impact on model performance. The authors also find no signs of overfitting from test set re-use on the popular SQuAD benchmark.

DE,DI *What Will it Take to Fix Benchmarking in Natural Language Understanding? [17]*. Survey paper with natural language processing that asserts learning problems should be well constructed, have adequate statistical power, and be representative of the task they aim to solve.

M *Translationese in Machine Translation Evaluation [52]*. The authors show that using "reverse"-direction sentences, which were translated from language A to language B but used for a B-to-A dataset, inflate human evaluation scores. They also examine prior claims of model-human parity and find evaluation problems such as not using a large enough test set; a re-evaluation suggests that the machine system was outperformed by humans.

DI *The Effect of Translationese in Machine Translation Test Sets [171]*. The inclusion of translationese, a translation artifact, in machine translation test sets inflates human evaluation scores for machine translation systems, and in some cases changes rankings of models.

DI *BERTs of a feather do not generalize together: Large variability in generalization across models with similar test set performance [93]*. Training the same NLP model architecture

(BERT) over a hundred different random seeds obtains consistent performance on MNLI, a natural language inference (NLI) dataset, but widely varying generalization performance, as measured on HANS, an NLI dataset that tests for biases learned on MNLI.

DE  *Probing Neural Network Comprehension of Natural Language Arguments [104]*. Finds that language models trained to solve a reasoning comprehension task exploit statistical cues within the dataset to achieve high performance.

M  *Re-evaluating the role of bleu in machine translation research [21]*. The authors highlight two situations where the use of BLEU fails to distinguish between translations which a human could tell apart and would rate differently. They find low correlation between BLEU scores and human judgements of adequacy and fluency.

B  *A call for clarity in reporting bleu scores [115]*. The most commonly used automatic metric (as opposed to human evaluation) in machine translation, BLEU, is not reported consistently: some papers preprocess text before scoring, and there are many parameters used by BLEU that aren't reported. The paper proposes a standarized tool for BLEU to solve these problems.

M  *BLEU might be Guilty but References are not Innocent [45]*. The authors show that improving reference translations improves correlation of BLEU with human judgement.

DI  *With Little Power Comes Great Responsibility [22]*. This paper describes the influence of statistical power in NLP experimental design and how small dataset size in GLUE make it difficult to distinguish between statistical noise and meaningful model improvements.

M, G  *Beyond Accuracy: Behavioral Testing of NLP models with CheckList [126]*. The authors propose an alternative evaluation paradigm to benchmarks, instead focusing on specific tests for known or anticipated failure modes for broadly relevant linguistic capabilities.

DE  *Learning the Difference that Makes a Difference with Counterfactually-Augmented Data [72]*. Crowdsourced perturbations of two NLP datasets cause model performance to drop, while learning on the regular and perturbed data improves on both domains and generalization to new domains. The two datasets are IMDB, a sentiment classification dataset [89], and SNLI [18], a natural language inference dataset.

B  *Fine-Tuning Pretrained Language Models: Weight Initializations, Data Orders, and Early Stopping [33]*. Finds that small factors such as random seed variance can have a huge impact on BERT performance, which is up to 7% on downstream tasks in the case of random seed variance.

M  *On The Evaluation of Machine Translation Systems Trained With Back-Translation [38]*. The authors show that BLEU fails to capture human preferences for models trained with back-translation.

DE  *Evaluating NLP Models via Contrast Sets [46]*. Benchmarks fail to address how NLP models perform in specific cases. The authors propose "contrast sets", where experts manually perturb test data points in a semantically meaningful way, to identify whether models are still able to output the correct answer. SOTA models perform worse on these contrast sets.

DI  *The Curse of Performance Instability in Analysis Datasets: Consequences, Source, and Suggestions [174]*. Analysis datasets are similar to standard benchmarks but specifically designed to test a linguistic capability or known failure mode. The authors find that model performance on benchmarks between random seeds is stable, but performance on analysis dataset can vary widely.

B  *On the State of the Art of Evaluation in Neural Language Models [95]*. The authors compare Recurrent Highway Networks (RHNs) against Long Short-Term Memory networks (LSTMs). They find that prior work demonstrating RHN superiority over LSTMs allocated more compute to RHNs, and find similar or competitive performance for LSTMs once this is controlled for.

DE  *Right for the Wrong Reasons: Diagnosing Syntactic Heuristics in Natural Language Inference [94]*. The Multi-genre Natural Language Inference dataset (MNLI) contains several examples of syntactic heuristics, where the answer can be predicted by following a simple rule, such as always predicting "contradiction" when the premise contains "not". The authors construct a new dataset HANS which contains examples that both satisfy and violate the heuristics, and show that SOTA models perform extremely badly (e.g. 100% to 0% accuracy) on the portion of HANS which violates the heuristics the models have learned from MNLI.

I *Revisiting Few-sample BERT Fine-tuning [172]*. Many previous papers have proposed solutions for stable BERT finetuning. The authors find that instability is caued by a bug in the ADAM implementation, and that fixing this bug reduces the advantage of propose finetuning methods.

B *It's Not Just Size That Matters: Small Language Models Are Also Few-Shot Learners [133]*. By reformulating SuperGLUE tasks into cloze-style questions, small language models can be can also be fine-tuned to have performace better than GPT-3.

DE *Robustness Gym: Unifying the NLP Evaluation Landscape [47]*. Presents software developer tools that cover a range of different NLP metrics, datasets, etc. in order to help practicioners evaluate their models in various conditions.

G, M *Utility is in the Eye of the User: A Critique of NLP Leaderboards [40]*. Authors highlight that leaderboards fail to measure (i.e., don't come with metrics for) factors beyond model performance, such as model size or inference speed or environmental impact.

DE *How Much Reading Does Reading Comprehension Require? A Critical Investigation of Popular Benchmarks [71]*. Reading comprehension benchmarks require models to pick the answer to a question given a passage. Models do well on reading comprehension benchmarks even when the passage or question is withheld, suggesting that the datasets are poorly constructed because knowledge of the passage or question is irrelevant.

## E.2    Computer vision (Image classification, object detection, Medical, General-purpose benchmarks, 3D shape reconstruction)

DE *AI for radiographic COVID-19 detection selects shortcuts over signal [30]*. Finds that models trained for COVID-19 detection through radiographs exploit spurious correlates that do not hold when deployed in different environments.

DE *Hidden Stratification Causes Clinically Meaningful Failures in Machine Learning for Medical Imaging [107]*. Finds that models trained for detecting a pneumothorax from chest x-rays latch onto obvious dataset-level heuristics such as the presence of a chest drain instead of adequately solving the task.

H *Evaluating Machine Accuracy on ImageNet [139]*. Humans are trained through documentation guidance and practice to classify objects in ImageNet and achieve comparable accuracy to modern machine learning models, though experience significantly less of a performance drop than models due to distribution shift. These labelers are about 3% to 9% better than the human performance levels reported from early 2015, indicating the variability of human baselines. Top-1 accuracy (a more natural task for humans) is almost perfectly linearly correlated with multi-label accuracy for the evaluated models, but humans fail more often for fine-grained categories (eg. differentiating dog breeds) while models fail more evenly across label categories.

B *Rethinking Few-Shot Image Classification: a Good Embedding Is All You Need? [151]*. Finds that the simple baseline of training a linear model on top of a supervised classifier in the context of meta-learning tasks can outperform a variety of previous meta-learning appraoches such as MAML.

M *Are we done with ImageNet?[12]*
The authors collect multi-label annotations for ImageNet via a modified crowdsourcing process. The results show a slightly plateauing trend, indicating that models may have overfit to specifics of the ImageNet distribution.

DE *Measuring Robustness to Natural Distribution Shifts in Image Classification [148]*. Finds that models trained on ImageNet fail to generalize well to other distributions with shifts in object pose, lighting, object composition, etc.

DE *In a forward direction: Analyzing distribution shifts in machine translation test sets over time [80]*. A fixed machine translation model scores better on newer machine translation test sets than older test sets (the Workshop on Machine Translation releases a new test set every year). The observed increase in scores for any single model is attributable to changes made in dataset construction which progressively removed translationese, a problematic translation artifact.

DE *Transfusion: Understanding Transfer Learning for Medical Imaging [117]*. This study looks at 2 medical image datasets: diabetic retinopathy prediction and chest x-ray prediction. They compare a few deep learning models and find that imagenet pretraining doesn't really help performance on these downstream datasets.

DE *CheXtransfer: Performance and Parameter Efficiency of ImageNet Models for Chest X-Ray Interpretation [73]*
The paper compares ImageNet performance of several CNN architectures to performance on X-ray classification. The authors find that X-ray classification performance has plateaued as a function of ImageNet performance, but ImageNet pre-training still helps on the X-ray dataset.

DE *Do Better ImageNet Models Transfer Better? [75]*
The authors evaluate ImageNet models on twelve other image classification datasets and find that better ImageNet models also perform better on the other datasetes, especially when the models are pre-trained on ImageNet.

DE *Is it Enough to Optimize CNN Architectures on ImageNet? [154]*. The authors train 500 ImageNet architectures on 8 other image classification datasets from different domains and find that the correlation between ImageNet performance and dowstream dataset performance varies wildly, with even negative correlations for some.

DE *Variable generalization performance of a deep learning model to detect pneumonia in chest radiographs: A cross-sectional study [167]*. Finds that models trained to diagnose pneumonia in chest radiographs in one hospital fail to generalize well to other hospital due to differences in data collection, equipment, patient populations, etc.

O *Does ImageNet Generalize to ImageNet? [124]*
The authors construct a new test set for ImageNet and find that overfitting from test set re-use did not occur despite a decade of competitive testing on this dataset. Instead, distribution shift led to a substantial drop in accuracy.

O *Does CIFAR-10 Generalize to CIFAR-10? [123]*
The authors construct a new test set for CIFAR-10 and find that overfitting from test set re-use did not occur despite a decade of competitive testing on this dataset. Instead, distribution shift led to a substantial drop in accuracy.

O *Cold Case: The Lost MNIST Digits* [163]
The authors construct a new test set for MNIST and find that overfitting from test set re-use did not occur despite two decades of competitive testing on this dataset.

B *A Baseline for Few-shot Image Classification [32]*. Finds that a simple transductively-tuned baseline can outperform all more complex methods (MAML, MetaOpt, etc.) on few-shot learning tasks when controlling for all other factors of variation.

B *What Do Single-view 3D Reconstruction Networks Learn? [149]*. Finds that simple baselines such as clustering and retrieval on top of the pretrained embedding space outperform recent deep methods for 3D reconstruction.

I *On Buggy Resizing Libraries and Surprising Subtleties in FID Calculation [112]*. The widely used Frechet Inception Distance (FID) metric for evaluating generative models is not consistently reported. Differences between image processing libraries and choices in implementing FID cause meaningful differences in scores.

DE *From ImageNet to Image Classification: Contextualizing Progress on Benchmarks [153]*. The authors provide multi-label annotations for ImageNet via a modified crowdsourcing process and study the impact of images with multiple labels on ImageNet accuracy metrics.

B *Overfitting in adversarially robust deep learning [127]*. The paper shows that early stopping combined with a simple loss function is competitive with more complicated loss functions that were proposed for adversarially robust image classification.

DI *Pervasive Label Errors in Test Sets Destabilize Machine Learning Benchmarks [105]*. Authors revealed significant label errors in mainstream datasets, such as an average error rate of 3.4% across the reviewed 10 datasets, including 6% of the ImageNet validation set.

### E.3 Meta-learning / Architecture search

B *Random search and reproducibility for neural architecture search [78]*. Given the same computational budget, random search with minor modifications (e.g. early stopping) outperforms state of the art neural architecture search methods.

B *Evaluating the Search Phase of Neural Architecture Search [166]*. Finds that random search within the penn treebank and cifar10 dataset search spaces leads to similar performance as leading neural architecture search algorithms when given equal compute.

### E.4 Generative models (GANs, generative language models)

M *HYPE: A Benchmark for Human eYe Perceptual Evaluation of Generative Models [173]*. This paper introduces a human benchmark for evaluation of generative models, which scores if a human can tell a real image vs fake. The authors found that HYPE scores were not correlated with commonly used automated metrics such as FID.

B *Are GANs Created Equal? A Large-Scale Study [88]*. Evaluates many GAN losses, fixing the backbone architecture, dataset, and other training details, and finds that most GAN models can reach similar performance given equal compute budget.

### E.5 Optimization for deep learning

B *Hyperband: a novel bandit-based approach to hyperparameter optimization [79]*. Finds that random search combined with early stopping outperforms more sophisticated Bayesian hyper-parameter optimization methods.

I *Decoupled Weight Decay Regularization [86]*. The authors point out that $L_2$ regularization is distinct from weight decay regularization for adaptive gradient algorithms like Adam, even though the former is often substituted for the latter. They show that implementing weight decay regularization improves Adam's generalization performance.

B *A Large Batch Optimizer Reality Check: Traditional, Generic Optimizers Suffice Across Batch Sizes [102]*. LARS and LAMB optimizers are designed to increase the speed of model training given large batch sizes. Traditional optimizers like Nesterov momentum and Adam perform comparably at large batch sizes, signifying that such interventions are not significant improvements when compared to an adequate baseline.

### E.6 Learning on graphs

B *Combining Label Propagation and Simple Models Out-performs Graph Neural Networks [67]*. Finds that incorporating a graph label propagation step with simple models outperforms more recent deep graph neural networks.

B *Benchmarking Graph Neural Networks [36]*. This paper documents common pitfalls and problems in benchmarking graph neural networks.

B *Simplifying Graph Convolutional Networks [162]*. Finds that a simple graph preprocessing step with an adjacency matrix combined with logistic regression outperforms more recent deep graph neural networks.

B *Pitfalls of Graph Neural Network Evaluation [140]*. Graph neural network papers (GNN) fail to control for relevant factors when making comparisons. The authors of this paper attempt to evaluate four GNN architectures while controlling for everything except architectures: keeping the optimizers, initialization methods, compute budget, etc, the same. Performance turns out to be similar between different GNNs.

### E.7 Tabular data & classical methods (ie. Medical (MIMIC))

B *Do we Need Hundreds of Classifiers to Solve Real World Classification Problems? [43]* When evaluating the performance of 179 classifiers on the whole UCI repository (121 data sets) [35], authors found that random forest classifiers outperformed any other type, with these methods achieving over 90% accuracy in 84.3% of the data sets.

B *Evaluating Progress on Machine Learning for Longitudinal Electronic Healthcare Data [10]*. Finds that on tabular data prediction tasks found on MIMIC-III, simple logistic

regression achieves comparable performance to more sophisticated methods developed over the past three years.

## E.8 Reinforcement Learning

DE, B *What Matters In On-Policy Reinforcement Learning? A Large-Scale Empirical Study [3].* Implements many RL algorithms and more than 50 code-level tricks and optimizations to consistently benchmark performance; one surprising finding is that policy initialization scheme plays a huge role in policy performance.

I *Implementation Matters in Deep Policy Gradients: A Case Study on PPO and TRPO [39].* The authors compare two deep policy gradient algorithms, Proximal Policy Optimization (PPO) and Trusted Region Policy Optimization (TRPO). They find that "code level optimizations", algorithmic modification described as auxillary details or undescribed altogether, are responsible for most of PPO's performance gain over TRPO and significantly affect algorithmic behaviour.

B *Deep reinforcement learning that matters [59].* Evaluation of reinforcement learning (RL) algorithms suffers from several issues: varying the random seed varies algorithm performance enough to change performance rankings; many under-reported hyperparameters greatly affect algorithm performance; different implementations of the same algorithm perform differently.

B *Simple random search provides a competitive approach to reinforcement learning [90].* A lightweight modification of random search achieves similar reward as SOTA reinforcement learning methods on MuJoCo Gym tasks while requiring fewer samples.

B *Towards Generalization and Simplicity in Continuous Control [118].* Simple methods using policies with linear and RBF parameterizations can solve many continuous control benchmarks, including MuJoCo Gym tasks. Further, the authors highlight that policies learned on the benchmarks are trajectory-centric: when these policies are perturbed, they fail to recover.

## E.9 Visual question answering

DE *Making the V in VQA Matter: Elevating the Role of Image Understanding in Visual Question Answering [49].* Finds that original VQA dataset is not balanced in terms of label distribution for certain questions, making achieving high performance relatively easy.

DE, T *On the Value of Out-of-Distribution Testing: An Example of Goodhart's Law [150].* Critiques the use of VQA-CP as a valid OOD dataset for VQA tasks, since VQA-CP inverts the VQA label distribution, and many robust methods explicitly rely on this fact.

## E.10 Information retrieval

B *Critically Examining the "Neural Hype": Weak Baselines and the Additivity of Effectiveness Gains from Neural Ranking Models [164].* Examines many information-retrieval papers from 2005-2019, and find that no approach (both neural or non-neural) comes close to the 2004 best.

## E.11 Metric Learning

B *A Metric Learning Reality Check [101].* Authors benchmark several deep metric learning algorithms on three datasets under identical training conditions and find that papers have drastically overstated improvements over classic methods.

B *Unbiased Evaluation of Deep Metric Learning Algorithms [41].* Authors benchmark several deep metric learning algorithms on three datasets under identical training conditions and find that older methods perform significantly better than previously believed.

B *Revisiting Training Strategies and Generalization Performance in Deep Metric Learning [129].* Authors benchmark several deep metric learning algorithms on three datasets under identical training conditions and find that generally, performance between criteria is much more similar than literature indicates.

## E.12 Recommender Systems

B *A Troubling Analysis of Reproducibility and Progress in Recommender Systems Research [25]*. This is a recommender systems reproducibility experiment, where they compare the proposed methods to a range of baselines on the datasets the original papers used; 11 of the 12 methods were outperformed by simple baselines on the datasets the respective paper had identified.

B *On the Difficulty of Evaluating Baselines: A Study on Recommender Systems[125]*. This is a recommender systems reproducibility experiment on the MovieLens-10M benchmark; finds that a well-tuned vanilla matrix factorization baseline significantly outperforms more recent methods reported in the literature.

## E.13 Semi-supervised /Unsupervised representation learning

B *Realistic Evaluation of Deep Semi-Supervised Learning Algorithms [108]*. This work is a standardized evaluation of semi-supervised learning algorithms on SVHN and CIFAR-10; they find that prior work underestimated the performance of fully supervised learning in the small-n regime and that ImageNet pre-training + fine-tuning with few samples does better than any of the semi-supervised methods they benchmarked.

## E.14 General / other

DE *Machine Learning that Matters [156]*. A scientist hoping to use machine learning for practical applications gets frustrated with the inadequate quality of the UCI repository [35] and the benchmark culture it perpetuates. She advocates instead for a more systems-level perspective on machine learning development and evaluation.

H *Performance vs. competence in human–machine comparisons [44]*. There is a difference between possessing human-level ability ("competence"), and the superficial demonstration of a skill ("performance"). Many models perform better than human counterparts on a given learning problem but do not achieve this performance in a human-like way, and thus fail to demonstrate competence when tested for that skill outside the scope of the initial learning problem.

DE,DI *A Flawed Dataset for Symbolic Equation Verification [28]*. A synthetic dataset for equation verification is heavily critiqued for the lack of rigor in how it is generated, the correctness of the axioms presented and the relevance of the task represented.

DI *"Everyone wants to do the model work, not the data work": Data Cascades in High-Stakes AI [132]*. Authors interview 53 ML practitioners in 6 countries and conclude that data work remains under-valued as a research topic of interest, even though data labelling consists of 25-60% of the cost of model development. They identify that data issues compound on each other in "data cascades", contributing to critical failures in model deployment within high stakes scenarios.

M *Accounting for variance in machine learning benchmarks [16]*. Several sources of variation in the dataset and implementation of machine learning models can obscure our understanding of their performance (eg. data sampling, data augmentation, parameter initialization, and hyperparameters choices). This paper recommends randomization and more robust trial reporting in order to appropriately and consistently address these issues.

G *Pitfalls in Machine Learning Research: Reexamining the Development Cycle [13]*. This paper comments on the challenges throughout the model development lifecycle that contributes to failures in machine learning deployment. Algorithmic design, data collection, and evaluation practices are named as concrete areas of concern - authors recommend interventions such as third party assessment, statistical testing and data audits.

B *Can You Trust Your Model's Uncertainty? [141]* This survey of uncertainty estimation methods shows that ensemble methods consistently outperform the rest.

O *The Ladder: A Reliable Leaderboard for Machine Learning Competitions [14]*
The paper introduces a specific attack on competition leaderboards demonstrating that overfitting from test set re-use is easily possible. The paper also describes a mechanism to protect from overfitting.

O *A Meta-Analysis of Overfitting in Machine Learning [128]*
The authors survey more than 100 classification competitions on Kaggle and find little to no overfitting from test set re-use.

DE *Underspecification Presents Challenges for Credibility in Modern Machine Learning [26]*.
Machine learning models that are identically trained and developed fail in different ways once deployed - this is partially due to the "underspecification" of the learning problem, in which features of the problem in the training domain are unaccounted for with respect to their influence on performance in the deployment domain.

DE,B *In the wild: From ml models to pragmatic ml systems [157]* Implicit assumptions in the experimental setup of few-shot and continual learning tasks obscure a clear understanding of performance measurement. The FLUID framework re-introduces certain experimental design considerations that need to be explicitly designed for real world model deployment.

DE,DI *Data and it's (dis)contents [113]*. The culture around dataset development, use, and distribution demonstrates a lack of cautious attention paid to this critical aspect of broader machine learning development.