# OpenReview forum: "Are We Learning Yet? A Meta Review of Evaluation Failures Across Machine Learning"
_NeurIPS.cc/2021/Track/Datasets_and_Benchmarks/Round2 — NeurIPS 2021 Datasets and Benchmarks Track (Round 2)_

### Official Review · Reviewer_qTGV · 2021-09-18
**Good framework for understanding model evaluation failures**

**Rating:** 9
**Confidence:** 4
**Correctness:** To my knowledge, yes.
**Clarity:** Yes, it was easy to read.

**Strengths:**

This is the first contribution I know of that does the following:
- Clearly define the difference between learning problems and tasks
- Identify evaluation failure types separately for learning problems and for tasks
- For each failure mode, identifies examples of these failures in the ML research literature

It will be a good reference for the ML research community about what failure modes to look out for and how to define their research's contribution (as either an advance on a learning problem or a task). The appendix also offers some guidance on how to avoid some failure modes.

The ethical implications of the paper are positive as it is providing guidance on how to improve research integrity. It will hopefully help to prevent some deployments of badly-performing AI that may have been evaluated inadequately.

**Weaknesses:**

N/A

**Additional Feedback:**

I have a minor comment about section 4.4.1 ("Reliance on simple heuristics").

I think the section title doesn't adequately describe the problem. Presumably, if a task could genuinely be solved with simple heuristics, that's not a problem (and maybe it's even good news). The problem that the authors describe in that section is rather that task performance was evaluated on inappropriate datasets, resulting in simple heuristics appearing to succeed when actually, when evaluated on appropriate datasets, they don't. Maybe the failure mode could be re-characterized as reliance on 'inappropriate' heuristics (something along those lines).

This raises a bigger issue about whether a task definition should include notions of what the 'appropriate' heuristics or datasets for it should be. I understand why the authors chose to leave out details like these from the definition of a task. However, it seems like specifying these details could eventually be important if we want to understand what, in the context of a specific task, counts as a "transfer" or a "distribution shift". I get that there's no room to address this in this paper, so I'm offering it as a suggestion for future work.

**Documentation:**

N/A

**Relation To Prior Work:**

In the appendix, the authors have a comprehensive review of prior work relating to evaluation. It's clear that their contribution is a meta-review of all this work that synthesizes lessons across all of them.

**Summary And Contributions:**

This paper presents a useful conceptual framework for understanding different types of model evaluation failures. The terminology and conceptual distinctions will be helpful in guiding the community's ideas of what evaluation should be.

---

> ### Author Response · Authors · 2021-09-27
> **Response to reviewer qTGV**
>
> We thank the reviewer for the feedback and kind words. We hope that our terminology and conceptual framework prove useful to the community.
>
> > I have a minor comment about section 4.4.1 ("Reliance on simple heuristics"). I think the section title doesn't adequately describe the problem. Presumably, if a task could genuinely be solved with simple heuristics, that's not a problem (and maybe it's even good news). The problem that the authors describe in that section is rather that task performance was evaluated on inappropriate datasets, resulting in simple heuristics appearing to succeed when actually, when evaluated on appropriate datasets, they don't. Maybe the failure mode could be re-characterized as reliance on 'inappropriate' heuristics (something along those lines).
>
> We agree with the reviewer that the delineation is between (i) when a task can be “genuinely” solved simply and (ii) when task performance is evaluated on inappropriate learning problems that allow for shortcut learning. This section was originally titled “Reliance on spurious features”, which suggested more strongly that the apparently simple heuristics disappear when evaluated on more representative learning problems. However, “spurious features” is also sometimes used in the context of distribution shift (Section 4.4.2), and we did not want to conflate these two notions. We will modify Section 4.4.1 to emphasize the intended framing and clarify the terminology.
>
> > This raises a bigger issue about whether a task definition should include notions of what the 'appropriate' heuristics or datasets for it should be. I understand why the authors chose to leave out details like these from the definition of a task. However, it seems like specifying these details could eventually be important if we want to understand what, in the context of a specific task, counts as a "transfer" or a "distribution shift". I get that there's no room to address this in this paper, so I'm offering it as a suggestion for future work.
>
> The reviewer raises a welcome point about how a more detailed definition of a “task” might restrict the learning problems which could be associated with each task. We debated this question in detail while writing the manuscript, and ultimately chose to forgo a more detailed definition of tasks for this version due to the already broad scope of the manuscript. This topic is an interesting direction for future research.

---

### Official Review · Reviewer_Yakz · 2021-09-20
**Limited contribution**

**Rating:** 6
**Confidence:** 4
**Correctness:** The claims are rather well-known and …
**Clarity:** The paper is clearly written.

**Strengths:**

- The paper raises important and common-sense conclusions on evaluating the performance of algorithms.
- The paper has the potential to ignite a stream of concrete steps in terms of more realistic evaluation metrics for ML models.
- The paper is nicely written.

**Weaknesses:**

- My main criticism is that the paper proposes common issues with ML evaluations, that are somehow well known by community experts. This raises some doubts about the impact of this paper.
- Secondly, the authors describe the issue of validity, but the manuscript falls short of giving concrete solutions to concrete problems.
- Thirdly, the paper is in large sections written with vague arguments with a limited scientific justification of claims. To give an illustration, sec 3.2 "test set size", when saying "use sample sizes too small to detect ...", what exactly is considered "too small", and how do you come up with that conventional term.
- Finally, the paper has some contradictory sections. While on one hand, the authors criticize the use of simple evaluations based on a single metric, on the other hand, they construct arguments based on single-metric results. E.g. In the paper that claims vanilla matrix factorization outperforms deep models, the results are compared using single-values too, concretely Recall, or NDCG. Same for other examples in 3.4.1.


**Additional Feedback:**

---- UPDATE ----

I raised the score after the authors' comments.

**Documentation:**

No documentation as no dataset is provided.

**Ethics:**

No issues I can assess.

**Relation To Prior Work:**

I am not well aware of a stream of research that resembles the manuscript, therefore, it is hard to judge how comprehensive is the coverage of prior work.

**Summary And Contributions:**

The paper introduces a critical perspective into the mechanism of evaluating machine learning methods. The authors highlight two modes of failure, in terms of internal and external classifications.

---

> ### Author Response · Authors · 2021-09-27
> **Response to reviewer Yakz (1/2)**
>
> We thank the reviewer for providing detailed feedback about the paper.
>
> > My main criticism is that the paper proposes common issues with ML evaluations, that are somehow well known by community experts. This raises some doubts about the impact of this paper.
>
> We agree that many of these failure modes may be known by community experts in specific machine learning sub-areas. However, we believe the long list of papers we reference speaks to the fact that machine learning researchers continue to stumble on these evaluation issues. To our knowledge, we know of no other prior work that catalogues, discusses, and ties together these failure modes across sub-areas within ML.
>
> Additionally, we believe our framework for discussing and evaluating external validity failures is a significant contribution. External validity failures are currently much more sparsely discussed, which is a concern since we ultimately care about transferring progress to downstream tasks. Our nomenclature around learning problems, tasks, and transfer can help the community formalize how we approach and represent external validity problems and evaluation challenges.
>
> > Secondly, the authors describe the issue of validity, but the manuscript falls short of giving concrete solutions to concrete problems.
>
> We provide a list of actionable recommendations for each failure mode in Appendix B. These recommendations are intended to be widely-applicable, but are not meant to solve every evaluation failure. We refrained from further highlighting the recommendations in the appendix for two reasons:
> First, no checklist can ever be complete enough for all possibilities. The range of how failure modes manifest is vast and a predetermined checklist is unlikely to win a game of whack-a-mole.
> Second, many of the failure modes remain open areas of research, such as overfitting from test set reuse, and sensitivity to real-world distribution shifts. For these failure modes, we do not have the right answers, and as a field machine learning has yet to discover all the right questions.
>
> [continues in [(2/2)](https://openreview.net/forum?id=mPducS1MsEK&noteId=Rnk-mQuTKGP)]

---

> ### Author Response · Authors · 2021-09-27
> **Response to reviewer Yakz (2/2)**
>
> [continues from [(1/2)](https://openreview.net/forum?id=mPducS1MsEK&noteId=fhyg5nzwyZl)]
>
> > Thirdly, the paper is in large sections written with vague arguments with a limited scientific justification of claims. To give an illustration, sec 3.2 "test set size", when saying "use sample sizes too small to detect ...", what exactly is considered "too small", and how do you come up with that conventional term.
>
> We did not detail the statistical methodology behind the notion of “test sets too small to detect differences between models” in the main text due to space constraints, but we will elaborate on this question in the appendix for the final version. The goal of our paper is not to give an introduction into the applied statistics methodology required for constructing test sets of appropriate size, but rather to emphasize the importance of this point and provide references that go into more depth. Indeed, the cited references for the aforementioned statement ([13, 19]) do go into detail about power analyses for tests, model performance variations, and the corresponding implications for test set size. We also provide a general recommendation Appendix B.2: “Optimize data test set size. Calculate what it means for the dataset to have a size that is statistically significant in reporting results.” It is unclear if the reviewer would like us to bring the analyses from the references into the main text or to further highlight the recommendation in the appendix, and so any clarification would be appreciated.
>
> We also welcome the opportunity to address any further uncertainties the reviewer has about vague language in the paper, if the reviewer would like to elaborate.
>
> > Finally, the paper has some contradictory sections. While on one hand, the authors criticize the use of simple evaluations based on a single metric, on the other hand, they construct arguments based on single-metric results. E.g. In the paper that claims vanilla matrix factorization outperforms deep models, the results are compared using single-values too, concretely Recall, or NDCG. Same for other examples in 3.4.1.
>
> This is a subtle point and we thank the reviewer for bringing it up. There is no contradiction between our statements about the use of solitary metrics and multiple metrics. Our position is that single metrics suffice for internal validity, but that external validity requires multiple learning problems (and hence multiple metrics).
>
> It is fair to say that vanilla matrix factorization outperforms deep models on a learning problem using only a single metric because comparing models on a learning problem means comparing them using that learning problem’s metric. All internal validity concerns are situated in the context of an isolated learning problem, meaning that internal validity is concerned with at most one metric. (Of course, it is fine to say that some method outperforms another method across several metrics - which is a statement about algorithm transfer, a subtype of external validity.) We will adjust the language in Section 3.4.1 to make this distinction more explicit.
>
> External validity deals with multiple learning problems, so statements about algorithm transfer or learning problem transfer become problematic when only a single metric (i.e., a single learning problem) is used to justify claims. For example, using a single metric on a single learning problem to claim performance on the broader task is often invalid, because it fails to provide insight on performance on other learning problems for that task.
>
> If the reviewer would identify other perceived contradictions, we would be happy to also address those.

---

### Official Review · Reviewer_rWrk · 2021-09-21
**Thorough analysis that could have more detail in the writeup**

**Rating:** 7
**Confidence:** 4
**Correctness:** The claims are correct and supported …
**Clarity:** Yes. The paper is well-written and mo…

**Strengths:**

There is no doubt the paper addresses an important topic with broad impact. The authors have done a lot of work surveying plenty of literature and organizing the results into a usable taxonomy. I especially appreciate the nuanced descriptions of failure modes which sometimes identify an “obvious” problem while also noting that serious issues related to that problem have not been observed yet. Interesting open problems and future directions are proposed throughout.

**Weaknesses:**

The paper is extensive, but still feels incomplete at times. Some descriptions of failure modes feel too brief and shallow, which is presumably due to space constraints. In addition, the taxonomy could be refined further. Some minor related comments are given in the additional feedback section.

**Additional Feedback:**

The descriptions of some failure modes could be clearer and/or more rigorous. I was sometimes relying primarily on the examples illustrating the failure mode to understand it’s definition better.

For 3.1 Algorithms, how do pressures from conferences to release code and consider reproducibility affect this failure mode?

Also 3.1, I would like to see more on the issue undocumented details. So many papers lack key details on every aspect of how experiments are conducted.

3.2 Label Errors is very brief. It may be worth discussing additional issues here such as how label errors relate to bias and fairness.

3.2 starting on Line 153: “Relatedly, [72] find that the removal of poor data from machine translation test sets over time has significantly contributed to the observed performance improvement in scores over the past decade.”
This seems like it should be a separate issue from “Contaminated Data” as it is more specifically related to the benchmark drifting and becoming easier over time.

Where does the following failure mode sit in your taxonomy? Sometimes algorithms perform so well on a benchmark that it appears the benchmark is just too easy (this can also lead to the inability to detect performance differences if multiple algorithms all do really well). One scenario where this occurs is when an old benchmark dataset is repurposed for a newly introduced problem variation. Does this fit somewhere in 3.4 Comparison to inadequate baselines? Or is it more similar to some test set size issue?

Could you speak more to the benefits and drawbacks of the choices made in this paper regarding how the taxonomy is structured (e.g., separating internal and external at the highest level)? In other words, what benefits and drawbacks come with the specific perspective of this work?

I would like to see Section 5 expanded if extra space is given to accepted papers.

I like Figure 2 a lot, but I don’t know if it justifies the space it occupies. Maybe it could just be smaller.


**Documentation:**

N/A

**Relation To Prior Work:**

Prior work is clearly discussed.

**Summary And Contributions:**

The authors perform a thorough meta-review to develop a taxonomy of failure modes for evaluating machine learning. They present a categorization and description of numerous failure modes along with many warnings and future directions.

---

> ### Author Response · Authors · 2021-09-27
> **Response to reviewer rWrk (1/2)**
>
> We would like to thank the reviewer for providing particularly detailed feedback.
>
> > Some descriptions of failure modes feel too brief and shallow, which is presumably due to space constraints.
>
> > The descriptions of some failure modes could be clearer and/or more rigorous. I was sometimes relying primarily on the examples illustrating the failure mode to understand its definition better.
>
> Space constraints are indeed a limitation here (we do elaborate on certain failure modes in Appendix B). If the reviewer felt that certain descriptions were too brief or confusing, it would be helpful to know which specific subsections the reviewer is referring to, so we can further elaborate in the text or add to our expanded descriptions in the appendix. Since there is the opportunity of adding an additional page of content to the main text, we would be happy to provide more details for the respective failure modes in the next version.
>
> > For 3.1 Algorithms, how do pressures from conferences to release code and consider reproducibility affect this failure mode?
>
> Overall we view code releases as a positive measure, so pressure from conferences to release code and data (when safe) is good for research. However, code releases are unfortunately not sufficient to fully address the problems described in Section “3.1 Algorithms”. A code release enables researchers to reproduce specific numbers or experimental results, but validity issues still persist if researchers are unaware that two implementations behave differently (for example, in the case of TRPO vs DDPG in Section 3.1 “Algorithms” or see Appendix B.1). Moreover, performance gains may still be incorrectly attributed even if code is released (described in Section “3.4.2 Controlling for algorithmic details”).  Instead of focusing solely on code releases and reproducibility, proper ablation studies are necessary to decompose performance gains. For algorithmic details, placing more emphasis on differences between code bases, or noting which code base was used, would begin to address this.
>
> > Also 3.1, I would like to see more on the issue undocumented details. So many papers lack key details on every aspect of how experiments are conducted.
>
> We elaborate further on Section 3.1 “Implementation variations” in Appendix B.1, though it seems the reviewer is asking more about underspecified experimental protocols. It would be helpful if the reviewer could provide some clarification on what is intended by “undocumented details” to ensure we fully address this point. We note that Section 3.4.2 “Controlling for Algorithmic Details” also elaborates on experimental details and discusses recent work seeking to control for factors like hyperparameter compute budget, optimizers, and network backbones.
>
> > 3.2 Label Errors is very brief. It may be worth discussing additional issues here such as how label errors relate to bias and fairness.
>
> We thank the reviewer for pointing this out and will use part of the additional page in the final version to address bias and fairness. Issues specific to label errors include offensive, incorrect or exclusionary labels [A,B,C,D]. For the final version, we hope to comment on how biased evaluations can further disguise model failures for under-represented population subgroups [E], and specifically how label errors and poor task framing can contribute to biased outcomes [F]. We would appreciate any additional references the reviewer may have in mind regarding these points.
>
> Finally, we briefly remark that bias and fairness can also arise as external validity issues, e.g., as measurement misalignments between an aggregate population and its constituent populations. We will clarify this in the final version of the paper.
>
> * [A] Scheuerman, Morgan Klaus, et al. "How We've Taught Algorithms to See Identity: Constructing Race and Gender in Image Databases for Facial Analysis." Proceedings of the ACM on Human-Computer Interaction 4.CSCW1 (2020): 1-35.
> * [B] Denton, Emily, et al. "Bringing the people back in: Contesting benchmark machine learning datasets." arXiv preprint arXiv:2007.07399 (2020).
> * [C] Prabhu, Vinay Uday, and Abeba Birhane. "Large image datasets: A pyrrhic win for computer vision?." arXiv preprint arXiv:2006.16923 (2020).
> * [D] Crawford, Kate, and Trevor Paglen. "Excavating AI: The politics of images in machine learning training sets." AI & SOCIETY (2021): 1-12.
> * [E] Raji, Inioluwa Deborah, and Joy Buolamwini. "Actionable auditing: Investigating the impact of publicly naming biased performance results of commercial ai products." Proceedings of the 2019 AAAI/ACM Conference on AI, Ethics, and Society. 2019.
> * [F] Mullainathan, Sendhil, and Ziad Obermeyer. "On the Inequity of Predicting A While Hoping for B." AEA Papers and Proceedings. Vol. 111. 2021.
>
> [continues in [(2/2)](https://openreview.net/forum?id=mPducS1MsEK&noteId=YPGcll9kYwP)]

---

> > ### Comment · Reviewer_rWrk · 2021-10-04
> > **Response to author response**
> >
> > Thank you for the thorough and thoughtful response. I have read it carefully and will address two points you asked for clarification on.
> >
> >
> > > Space constraints are indeed a limitation here (we do elaborate on certain failure modes in Appendix B). If the reviewer felt that certain descriptions were too brief or confusing, it would be helpful to know which specific subsections the reviewer is referring to, so we can further elaborate in the text or add to our expanded descriptions in the appendix. Since there is the opportunity of adding an additional page of content to the main text, we would be happy to provide more details for the respective failure modes in the next version.
> >
> > Unfortunately, I didn’t note each specific confusing section as I was reading. I’ll try note some examples here.
> >
> > One example is “Contaminated data”  in 3.2 which confused me a bit as evidenced by my question about line 153. This topic starts with a specific example of contamination that could lead to overestimation, then it just jumps to another different example of contamination. There is no definition of contaminated data or explanation of what these two examples are illustrating and how they complement each other. Adding a sentence or two before, in between, and/or after these two examples would help.
> >
> > The topics in 3.1 had some similar problems to a lesser extent in “Algorithms” and “Metrics”.
> >
> > By contrast, I’d say “Test set size” in 3.2 has a nice, succinct definition of the issue.
> >
> >
> > > We elaborate further on Section 3.1 “Implementation variations” in Appendix B.1, though it seems the reviewer is asking more about underspecified experimental protocols. It would be helpful if the reviewer could provide some clarification on what is intended by “undocumented details” to ensure we fully address this point. We note that Section 3.4.2 “Controlling for Algorithmic Details” also elaborates on experimental details and discusses recent work seeking to control for factors like hyperparameter compute budget, optimizers, and network backbones.
> >
> > I was referring to both implementation variations as you define them and underspecified experiment protocols with the latter being outside the scope of Section 3.1. I see that Appendix B.1 does elaborate more on some of this.
> >
> >
> >
> > In revisiting the paper, I noticed “alogorithm” on line 114.

---

> ### Author Response · Authors · 2021-09-27
> **Response to reviewer rWrk (2/2)**
>
> [continued from [(1/2)](https://openreview.net/forum?id=mPducS1MsEK&noteId=HIK6s0nffEy)]
>
> > 3.2 starting on Line 153: “Relatedly, [72] find that the removal of poor data from machine translation test sets over time has significantly contributed to the observed performance improvement in scores over the past decade.” This seems like it should be a separate issue from “Contaminated Data” as it is more specifically related to the benchmark drifting and becoming easier over time.
>
> We used this example to show that contaminated data can cause underestimation of model performance in addition to overestimation, as shown in the previous example starting on line 152. In this case, datasets with less contaminated data showed higher model performance. As written, the paragraph over-emphasizes the temporal aspect, which is not necessary to make the point about underestimation of model performance. We will change the writing in the final version to avoid confusion.
>
> > Where does the following failure mode sit in your taxonomy? Sometimes algorithms perform so well on a benchmark that it appears the benchmark is just too easy (this can also lead to the inability to detect performance differences if multiple algorithms all do really well). One scenario where this occurs is when an old benchmark dataset is repurposed for a newly introduced problem variation. Does this fit somewhere in 3.4 Comparison to inadequate baselines? Or is it more similar to some test set size issue?
>
> This failure mode is an external validity issue. If the only issue with the benchmark is that it is “too easy” because all models perform identically, the benchmark is internally valid, but causes complications when trying to extrapolate performance to other learning problems. In other words, a benchmark that is “too easy” instantiates the underlying task poorly because it is “too easy” in relation to other learning problems for that task.
>
> While the reviewer’s main concern seems to be the case when a benchmark fails to capture the full difficulty of a task, the illustrated situation could overlap with test set size issues. If the models all perform so similarly that it becomes statistically impossible to distinguish between their performance, then the test set is too small. We note that this can occur at any point in the performance spectrum: high-performing models can be clustered this way just as low-performing ones can be.
>
> > Could you speak more to the benefits and drawbacks of the choices made in this paper regarding how the taxonomy is structured (e.g., separating internal and external at the highest level)? In other words, what benefits and drawbacks come with the specific perspective of this work?
>
> We see separating internal and external validity as a useful tool to reason about ML evaluations. Internal validity failures are currently commonly discussed in the literature across a variety of ML subfields (see Appendix C). External validity failures, however, are much more sparsely discussed. We wanted to highlight external validity issues as important considerations that need to be consistently addressed, since ultimately machine learning cares about transferring progress to downstream tasks. Our nomenclature around learning problems and tasks can help the community formalize how we approach and represent external validity problems and evaluation challenges.
>
> One potential drawback of our approach lies in our focus on research validity, in contrast to the surrounding moral values and goals of research agendas. Too narrow a focus on the technical integrity of research can obscure consequential questions around the ethical value of research (e.g., an ethically questionable research direction remains ethically questionable even if the research is of high technical quality). We hope that our focus on validity issues in machine learning research does not crowd out other reflective questions about ML research. Ideally there is an interplay between these considerations: questions of fairness, privacy, trustworthy machine learning, etc. all must rest on solid technical foundations for the associated statements to be meaningful.
>
> > I would like to see Section 5 expanded if extra space is given to accepted papers.
>
> Accepted papers will indeed be given an additional page, so we will expand this section in the final version.

---

### Official Review · Reviewer_yniY · 2021-09-21
**This paper looks like simple common sense, except it is not simple nor common**

**Rating:** 7
**Confidence:** 5
**Correctness:** It is hard to accept that the taxonom…
**Clarity:** The paper is very clearly written.

**Strengths:**

- The main strengths of the paper are:
    - taxonomy of benchmark evaluation failure modes
    - extensive summary and classification of related work with those failure modes
    - can be used as a checklist of issues to avoid and, if followed, has the potencial to have large, positive impact in the community.

**Weaknesses:**

- The main weaknesses of the paper are:
    - The taxonomy is not complete. For example, a common issue is label leakage (although it could be argued that it fits under "errors in test set construction", none of the three examples there relate to label leakage) and another common issue is incorrect "test setup" (e.g., doing cross-validation on a time-sensitive dataset).
    - The paper "appears" too simple and obvious. But it is not!

**Additional Feedback:**

No additional feedback

**Documentation:**

The work is well documented, with supplementary material and a Github site

**Ethics:**

No ethics concerns.

**Relation To Prior Work:**

The relation with prior work is clear and a very exhaustive list of related work is mentioned.

**Summary And Contributions:**

The paper presents a taxonomy of typical failure modes (e.g., implementation variations, incorrect baselines, human baselines, etc) when producing benchmark studies, classifies those failure modes, provides descriptions for each one of the failure modes, and lists and summarizes extensive related work (in supplemental material) of more than one hundred other references as examples of those failure modes. While one would expect that all listed failure modes would be "obvious" for practitioners, the extensive list of related work with benchmark issues shows that, in practice, these failure modes are not obvious.

---

> ### Author Response · Authors · 2021-09-27
> **Response to reviewer yniY**
>
> We would like to thank the reviewer for the positive feedback. We agree that these issues are not as simple or obvious as they may appear at first glance.
>
> > The taxonomy is not complete. For example, a common issue is label leakage (although it could be argued that it fits under "errors in test set construction", none of the three examples there relate to label leakage) and another common issue is incorrect "test setup" (e.g., doing cross-validation on a time-sensitive dataset).
>
> > It is hard to accept that the taxonomy is complete/correct.
>
> Thank you for bringing up these examples of evaluation failures. Label leakage indeed belongs in the taxonomy under “errors in test set construction”. We will include this example in the final version of our paper. Similarly, the “errors in test set construction” section will also mention that test set construction with time-sensitive data requires extra care. We would appreciate any references the reviewer may have in mind regarding these issues.
>
> Regarding completeness, we would like to distinguish between two notions of completeness here:
>
> A) Whether we list all examples of concrete failure modes for the categories in our taxonomy
>
> B) Whether the categories in our taxonomy cover all failure modes
>
> We agree that our manuscript is not complete in sense A), but emphasize that this was never the goal of our work. Some failure mode categories such as “implementation variations” encompass a myriad of different concrete examples that are hard or impossible to list exhaustively. However, we indicate enough examples to illustrate what we mean by this type of failure.
>
> Regarding completeness notion B), the question is complicated by the fact that there are many different forms of machine learning (supervised, unsupervised, reinforcement learning, etc.). Our taxonomy is certainly not complete for reinforcement learning (e.g., we do not discuss issues pertaining to simulation environments), but this was again not the goal of our work. Instead our goal was to focus on i.i.d. supervised learning as this is the setting most commonly studied in machine learning. When scoped to the context of supervised learning, we do believe that our taxonomy is complete in describing the major categories of failures, given the benchmark paradigm of machine learning evaluations. Our rationale is based on the perspective of the benchmark paradigm in machine learning research pictured in Figure 1. Each step in the pipeline (“Trained model”, “Test set performance”, etc) is associated with failure modes at that stage (“Implementation variations”, “Test set construction”, etc). An evaluation failure must occur at one of the stages in the evaluation pipeline, so by construction it falls into one of the existing failure modes. We will formalize this argument in the appendix for the final version and clarify that our taxonomy is complete only for the supervised learning setting. If the reviewer disagrees with this assessment, we would find it helpful if the reviewer could provide a concrete example of a failure mode in i.i.d. supervised learning that does not fall into our taxonomy. We would also be happy to clarify this point further in further comments on this platform.

---

### Decision · Program_Chairs · 2021-10-09

**Decision:**

Accept

**Comment:**

This paper provides a useful taxonomy and list of seemingly obvious failures in ML evaluations. After the author discussion period, all authors gave acceptance scores, and I agree with this suggestion. I encourage the authors to address the criticisms given in the individual reviews, using appendices for any details they did not spell out due to space constraints or other details they would like to add based on the reviewer comments.